# Online legal driving behavior monitoring for self-driving vehicles

Wenhao Yu[1,5], Chengxiang Zhao[2,5], Hong Wang [1] ✉, Jiaxin Liu[1], Xiaohan Ma[2], Yingkai Yang[1], Jun Li[1], Weida Wang[2], Xiaosong Hu [3] ✉ & Ding Zhao[4]

Defined traffic laws must be respected by all vehicles when driving on the road, including self-driving vehicles without human drivers. Nevertheless, the ambiguity of human-oriented traffic laws, particularly compliance thresholds, poses a significant challenge to the implementation of regulations on self-driving vehicles, especially in detecting illegal driving behaviors. To address these challenges, here we present a trigger-based hierarchical online monitor for self-assessment of driving behavior, which aims to improve the rationality and real-time performance of the monitoring results. Furthermore, the general principle to determine the ambiguous compliance threshold based on real driving behaviors is proposed, and the specific outcomes and sensitivity of the compliance threshold selection are analyzed. In this work, the effectiveness and real-time capability of the online monitor were verified using both Chinese human driving behavior datasets and real vehicle field tests, indicating the potential for implementing regulations in self-driving vehicles for online monitoring.

Legal driving is a prerequisite for the widespread adoption of self-driving vehicles (SVs) to ensure the safety of future transportation[1]. Independent online monitoring of the driving behavior of SVs is not only an essential means for government regulation of autonomous driving, such as provides substantial evidence for the traceability of traffic incidents, but also can provide warnings of violations to autonomous driving algorithms, helping improve their compliance with regulations. Currently, human-oriented traffic laws contain numerous ambiguous expressions, leading to varying interpretations by enterprises, and consequently, divergent behaviors in self-driving vehicle (SV) systems[2–4]. The implementation of human-oriented traffic laws for real-time vehicle driving remains a challenge. Until now, independent online legal driving behavior monitoring that seamlessly integrates into AVs and covers the entire road network is still lacking considering the rational and machine-interpretable compliance thresholds.

In the field of autonomous driving, most regulation-related research begins with the formalization of regulations. The teams of

Althoff and Bin-Nun made major pioneering contributions to the formalization of traffic laws and subsequent applications. In early studies, simple logic formulas were used to formalize traffic laws, such as first-order logic[5], deontic-order logic[6,7], and high-order logic[8,9]. However, these methods cannot describe the sequential nature of the typical driving behavior of traffic laws[10]. Recently, temporal logic-based methods such as linear temporal logic (LTL)[11] have gained traction because of their expressiveness in traffic-law representation. Extensions such as signal temporal logic (STL)[12,13] are additionally equipped with legality robustness degree, and metric temporal logic (MTL)[14–16] can further specify intervals for property fulfillment. Censi et al. proposed the concept of a hierarchical model for traffic law violation by implementing liability, ethics and culture-aware behavior specification as Rulebooks[17]. The primary applications of formalized traffic laws include monitoring, control synthesis, and formal verification[18]. Monitoring refers to checking the current or recorded driving behaviors of SVs violate traffic laws[11,19]. The control synthesis aims to solve a vehicle

[1]School of Vehicle and Mobility, Tsinghua University, 100084 Beijing, China. [2]School of Mechanical Engineering, Beijing Institute of Technology, 100081 Beijing, China. [3]Department of Mechanical and Vehicle Engineering, Chongqing University, 400044 Chongqing, China. [4]Department of Mechanical Engineering, Carnegie Mellon University, Pittsburgh 15213 PA, USA. [5]These authors contributed equally: Wenhao Yu and Chengxiang Zhao. ✉e-mail: hong_wang@tsinghua.edu.cn; xiaosonghu@ieee.org

controller to plan an optimal trajectory within traffic laws restrictions[20–23]. Specifically, Xiao et al. implemented the traffic laws into an online real-time planner by specifying their priorities by constructing a priority structure called Total Order over eQuivalence classes (TORQ)[21]. Formal verification aims to theoretically prove or ensure the legality of all possible behaviors of a given SV system[24–26]. Most applications focus on enhancing the compliance of SVs within the current traffic-law framework, and only a few are dedicated to offering independent and reliable sources of compliance monitoring data for government regulation and enterprise analysis. The latter goal requires that the monitor encompasses all traffic-law articles, operates continuously to cover all road sections without interfering with the SV system's decisions, and provides rational judgments based on the genuine driving behaviors of vehicles.

The understanding of ambiguous traffic laws varies significantly among individuals, and the key challenge in achieving rational judgments is the selection of thresholds for ambiguous articles, such as that the vehicle behind shall overtake from the left side of the vehicle in front after making sure that there is sufficient safe space, and that the vehicles making a turn may not interfere the vehicles and pedestrians that are let go straight forward. Many researchers have attempted to select thresholds for ambiguous articles, and some researchers[27] sought relevant guidance from suggestive documents, such as driver guides[28,29]. These suggestions are often derived from previous driving experience. The prevailing approach for threshold analysis is based on

theoretical models that specify thresholds using pre-designed models with kinematic principles. For example, the driver reaction model with the maximum brake distance[11,12,30,31] or the set-base prediction model[32–34], can be used to specify the safe distance. Thresholds from models are usually conservative owing to strict constraints that are applicable for decision-making to ensure legality. Recently, studies performed by Belta et al.[35,36] demonstrated the potential of constructing data-based models from datasets. Additionally, Bogdoll et al.[37] attempted to determine the threshold for the following distance based on the Waymo dataset, which accommodates variations in the behaviors of different traffic participants. However, the driving guidance cannot be utilized to determine whether a violation has occurred. Moreover, safety models tend to be conservative, making it difficult to blame aggressive drivers for not adhering to the safety model unless an incident has occurred. Thus, it is imperative to establish rational thresholds based on driving behavior to ensure that they align with the distribution of the majority of human drivers' behavior.

The focus of this study was to develop a fact-based online legal driving behavior monitoring system with the primary purpose of providing comprehensive, authentic feedback data for government regulations and violation alerts to improve traffic-law compliance (Fig. 1). Our approach focuses on creating a trigger-based hierarchical online monitoring architecture that is compliant with the semantic types of the traffic law and selecting compliance thresholds that are

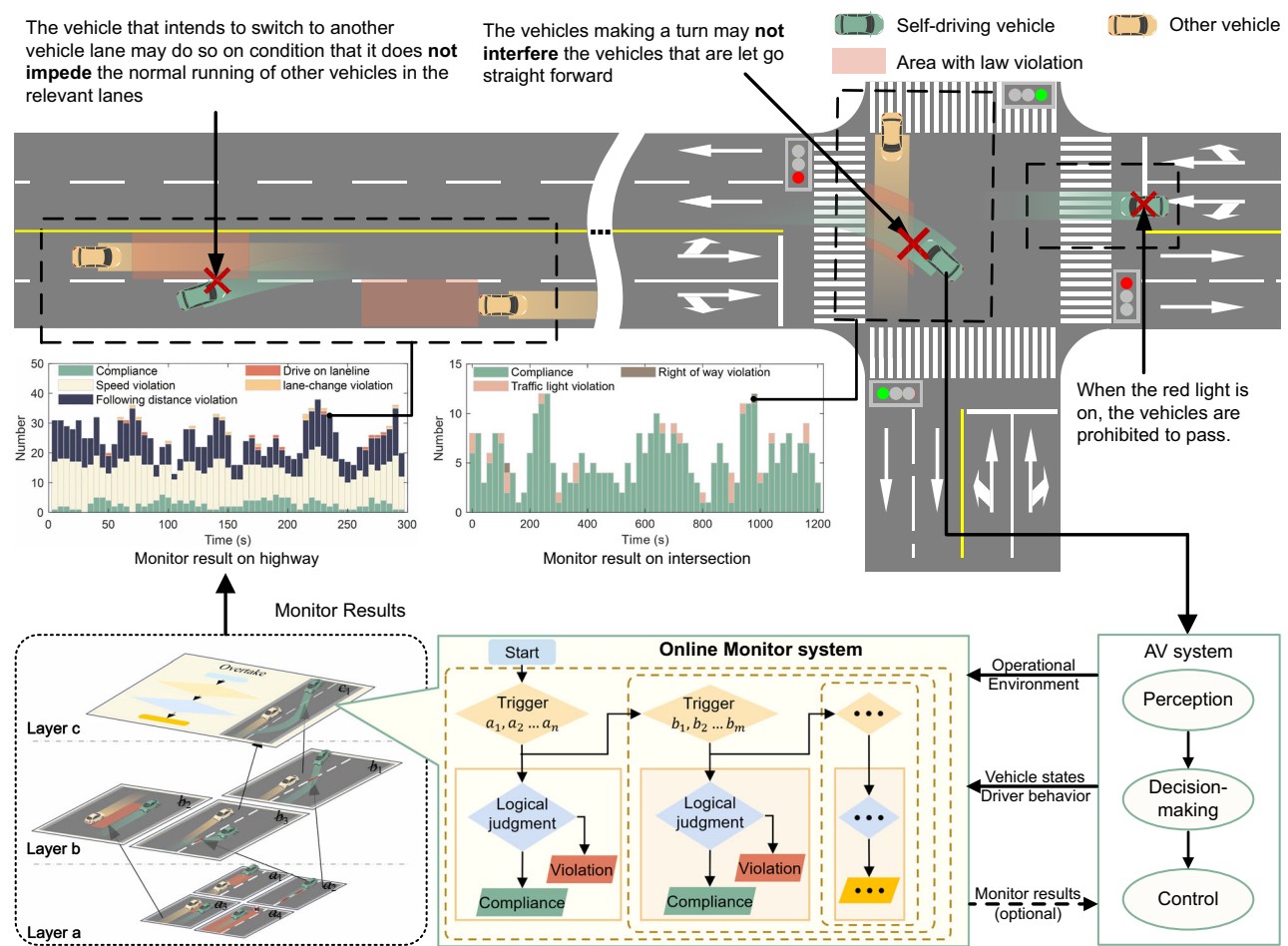

**Fig. 1 | Online traffic-law violation monitor for AVs.** This monitoring system is capable of deployment within the SV and monitors the SV's adherence to traffic laws. It receives real-time data from the SV system and provides continuous monitoring results of the ego vehicle. The monitoring system has a trigger-based hierarchical architecture that ensures structural integrity (e.g., drive on lane line $(a_2) \subseteq$ make lane-change $(b_1 \& b_3) \subseteq$ overtake $(c_1)$ or keep lateral distance$(a_3) \subseteq$ encounter $(b_2)$), which enhances the rationality of the monitoring results and simplifies maintenance in later stages.

consistent with the distribution of real human driving behavior. This distinguishes our monitor from existing traffic-law compliance verifiers and trajectory safety checkers. While the latter are mainly designed to validate trajectory compliance with specific traffic-law articles or to make safety-related decisions that require prediction and decision-making data, our monitor is dedicated to comprehensive traffic law compliance judgment utilizing only driving behavioral data. When applied in the open environment, other methods may encounter challenges in triggering correct law article monitoring in specific scenarios or in aligning with the behavior of the majority of traffic participants. This work has three key features that address these concerns.

(1) Trigger-based hierarchical online monitor architecture. This allows the continuous differentiation of the operational environment and behaviors of the surrounding traffic participants encountered by the ego vehicle, facilitating the correct monitoring of relevant law articles.
(2) Fact-based logical judgment and data-based thresholds. By analyzing the behavior of most traffic participants, compliance thresholds aligned with real-world behavior are obtained.
(3) Sensitivity analysis of thresholds. Through a sensitivity analysis, compliance monitoring thresholds are fine-tuned to strike a balance between false negatives (non-compliance but judged as compliance) and false positives (compliance but judged as non-compliance).

Erroneous triggers for compliance assessments in online monitoring can lead to inaccurate and misleading results. For example, if the monitoring system incorrectly triggers left-turn violation monitoring when the vehicle makes a left turn without entering or exiting a designated no left-turn zone, it can lead to conflicting internal information within the SV system; if this result is taken into the consideration of decision-making system, this will potentially result in unexpected behaviors. These outcomes are unacceptable to government authorities and enterprises. The challenge arises from the environmental specificity of traffic laws that typically regulate vehicle behavior within operational environments. To address this, each traffic-law article requires a detailed breakdown of its operational environment specifications and behavioral judgments. In conjunction with historical and real-time vehicle state information, operational environment specifications are designed as tailored trigger conditions for each corresponding legal article, and computable minimum atomic propositions are proposed, along with the formulation of input information requirements for the autonomous driving system (Fig. 1). Trigger-based hierarchical online monitoring allows real-time determination of the regulations that need to be monitored in the current scenario. This triggers only the articles that should be monitored to, increase the accuracy and rationality of monitoring. After all the articles are systematically organized, a hierarchical architecture based on common trigger relationships between regulations can further enhance the real-time performance. It also facilitates program modification, addition, and maintenance following subsequent traffic-law adjustments.

Traffic laws may explicitly state precise constraint values, such as maximum speed limitations or requirements to stop at red lights, leading to consistent formalization expressions and fewer controversies over judgment outcomes. However, in cases where traffic laws lack clarity and are ambiguously expressed, establishing rational judgment thresholds during the formalization process presents a challenge. While thresholds not explicitly designed for government regulation have exhibited positive impacts on compliance and safety in self-driving, persuading individuals who are considered to be in violation—particularly when penalties are involved—can be challenging if these thresholds do not align with the behavior of the majority of drivers. The thresholds intended for government oversight should be

conservative and align more closely with driver behavior. To address these challenges, we propose two general principles to determine the ambiguous compliance thresholds.

1. "No Crashes": For safety, there should be no crashes with other vehicles. This is ensured through safety-related indices such as the Time to Collision (TTC), Responsibility Sensitive Safety (RSS) and other kinematics models.
2. "No Changes": The ego vehicle's behavior should not be the reason to cause the change of other vehicles' behavior, which can be established by assessing nearby vehicles whose trajectories intersect with the ego vehicle. If no significant braking or steering responses are observed, we assume that the surrounding vehicle is not affected by the ego vehicle.

By applying these principles to real-world datasets, we can effectively identify parameter distributions related to unsafe and interfering behaviors. This helps to determine the range of compliance thresholds. Owing to the numerous factors that influence the behavior of each traffic participant, even when mathematical combinations of key factors are employed, eliminating the impact of other factors remains difficult. Therefore, establishing a sound threshold based on the information perceptible to the ego vehicle to effectively differentiate between compliant and non-compliant behaviors is a challenge. Sensitivity analysis is essential for determining the trade-off between false negatives and false positives within a given compliance-threshold range. Considering that the online monitoring system is designed to enforce government regulations, a lower tolerance for false-positive errors is preferred. During the data analysis process for selecting suitable thresholds, we evaluated the false-positive rates and the corresponding false-negative rates, ultimately establishing the final threshold values based on the cost function. Building on the aforementioned online monitoring system, we utilized Chinese-specific highway and intersection datasets to verify the feasibility of the system. The results indicated that the proposed online monitoring system effectively triggered the monitoring of corresponding articles and correctly produced violation judgments.

To evaluate the effectiveness of our approach and its real-time capabilities, the proposed online legal driving behavior monitor was deployed on an industrial personal computer (IPC) and then integrated into the FAW Jiefang commercial SV. Our online monitor receives the required information from the SV system to monitor traffic-law violations and then provides feedback to the SV system. Three typical traffic-law violation scenarios based on highway field test conditions were considered. The results indicated that under real-world conditions, our online monitoring system can consistently provide stable and accurate outputs with a computation time of approximately 1.3 ms. The system is designed for seamless integration with various SV systems as long as the required information is supplied; thus, it can not only be used for government supervision but also can serve as a traffic-law compliance advisor by incorporating prediction and decision data.

## Results
### Traffic law articles
Countries and regions around the world have their own distinct road traffic laws shaped by local cultures, histories, and social backgrounds. Although there are variations in laws among countries, these differences primarily lie in the thresholds of constraints on different behaviors. The behaviors constrained by the traffic laws and the meanings of the constraints are generally consistent. This consistency makes it possible to utilize a formalized framework of laws to solve the formalization problem of different laws in different regions. The specific analyses of the traffic laws can be found in the Supplementary Method.

Traffic laws in most countries and regions limit driving behavior primarily through the following four aspects: vehicle speed, distance,

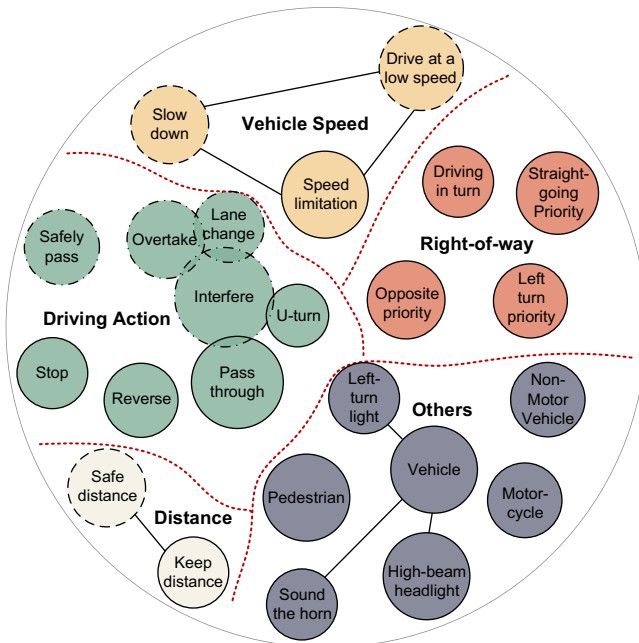

**Fig. 2 | Classification of traffic laws.** Each circle represents a different class of traffic-law constraint. The size of the circle represents the proportion of the class, and overlap indicates that both constraint classes appear in the same article. The solid edge line type of the circle indicates that there is no ambiguous expression involved, whereas the dash-dot line indicates that an ambiguous parameter is involved for certain articles and the dotted line indicates that the corresponding expressions in each article need to be clarified.

driving action, and right-of-way as shown in Fig. 2. Consider the Regulation on the Implementation of the Law of the People's Republic of China on Road Traffic Safety[38] as an example. In Chapter 4 (Road Traffic Regulations), there are 49 traffic articles, but only 25 are related to motor vehicle driving behaviors. These 25 articles are subdivided into 93 articles, including 23 articles related to speed limits, 7 articles related to distance restrictions, 33 articles related to maneuvering restrictions, and 18 articles related to right-of-way. The detailed classification is shown in Fig. 2.

Six representative articles were selected that satisfied the requirements. i.e., they (1) included all constraint types, (2) involved both precise and ambiguous thresholds, (3) involved real-time and continuous state constraints, and 4) could be analyzed and validated using the available datasets. Four articles on highway regulations were selected according to these requirements.

Article 44: The vehicle that intends to switch to another vehicle lane may do so on condition that it does not impede the normal running of other vehicles in the relevant lanes.

Article 78: When running on highway …, where there are two vehicle lanes in the same direction, the minimum speed for the left lane is 100 kilometers per hour…, where there is any discrepancy between the speed indicated by a speed limit sign put up on a road and the driving speeds mentioned above, a motor vehicle shall be driven at the speed indicated by the speed limit sign on the road.

Article 80: … when the speed is lower than 100 kilometers per hour, the distance from the vehicle in front may be narrowed appropriately, but the minimum distance may not be less than 50 meters.

Article 82.6: When driving a motor vehicle on the highway, the driver shall not drive over or on the dividing line of vehicle lanes or on the shoulder.

The thresholds for the speed limit and following distance are specified precisely. However, the not impede and drive over or on the dividing line behaviors are ambiguous and need to be defined.

Two key articles on intersection regulations were selected.

Article 38.1: When the green light is on, the vehicles are allowed to pass; when the yellow light is on, the vehicles that have gone beyond the stop line may continue to pass; when the red light is on, the vehicles are prohibited to pass.

Article 38.2: The vehicles making a turn may not interfere the vehicles that are let go straight forward.

Article 38.1 involves both real-time and continuous state behavior constraints, which are precisely defined. However, the not interfere behavior regarding right-of-way is ambiguous.

## Datasets

We utilized two datasets for verification: The AD4CHE (Aerial Dataset for China Congested Highway and Expressway) dataset[39] and the SIND (Signalized INtersection Dataset) dataset[40]. AD4CHE lasted approximately 307 mins, covering a trajectory length of 6540.7 km and encompassing 53,761 trajectories, including 68 records captured on four Chinese highways. Compared with the HighD dataset[41], AD4CHE covers intricate road structures, including curved roads, on/off ramps, multiple lanes, and various traffic flow states, as shown in Fig. 3(a–d), with abundant vehicle coordinate system parameters. In addition, we collected vehicle trajectories and traffic signal status information from four urban intersections to create the SIND dataset. This dataset encompassed four two-phase signalized intersections situated in different Chinese cities spanning 2300 km. The SIND dataset lasted for approximately 957 mins, encompassing 30,953 trajectories, including 53 records and involving 7 types of traffic participants (cars, buses, trucks, motorcycles, bicycles, tricycles, and pedestrians), as shown in Fig. 3(e–h). All available data were utilized to validate the online monitor. For further details, please refer to the source data.

## Threshold analysis

According to the selected articles, there are four thresholds in ambiguous expressions that need to be determined: the maximum allowable time of driving on the lane line ($t_{cl\_max}$) was determined to specify the expression drive over or on the dividing line in Article 82.6; when making lane-change, the minimum allowable TTC with the preceding vehicle ($TTC_{cl\_min}$) and the minimum allowable distance from the rear vehicle in the target lane ($d_{cl\_min}$) were determined to specify the expression not impede in Article 44; and the minimum allowable time difference between a left-turn vehicle and a straight-moving vehicle to the intersection point ($TTI_{diff\_min}$) was determined to specify the expression not interfere in Article 38.2. The last three thresholds relate to the risk of collision with other vehicles, so in the analysis, we considered the general principle of "No Crashes" and "No Changes".

The maximum allowable time of driving on the lane line $t_{cl\_max}$ was determined utilizing the AD4CHE dataset with the "No Crashes" principle. Because of the fact-based monitoring, only vehicle behavior data can be utilized, the trigger condition for monitoring is that the ego vehicle's bounding box overlaps with the lane lines, this process is considered a lane-change process in this article. Similar to the lane change maneuver (LCM) defined in the regulation UN ECE R157[42], as shown in Fig. 4a. In the dataset, 3510 instances were recorded in which a vehicle implemented a lane-change. Among them, 1753 instances involved complete lane-change trajectories. According to the collected data, the duration of all types of vehicles crossing lane lines during a lane-change was statistically analyzed, and the statistical results are shown in Fig. 4c. The statistical results followed an inverse Gaussian distribution with fitted parameters of $\mu = 2.791$ and $\lambda = 20.689$. Vehicles crossed lane lines for durations up to 6 s in 99.04% of the cases. Therefore, $t_{cl\_max}$ was determined to be 6 s, ensuring that standard lane-change maneuvers occurred within this specified time. This result differs from that in regulation UN ECE R79[43], as $t_{cl\_max}$ is defined as the time when a vehicle overlaps the lane lines, rather than the entire lane-changing duration.

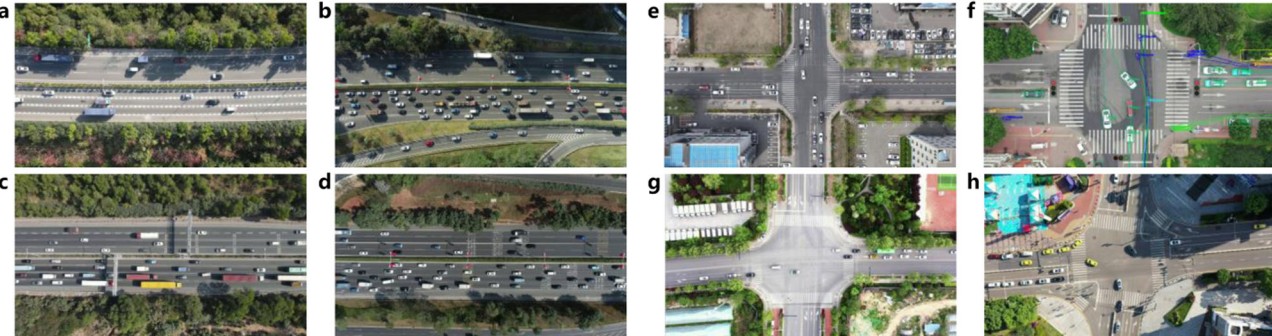

**Fig. 3 | Illustration figures of the Datasets.** Road structure in AD4CHE[39]. **a** Curved road. **b** Curved road with export and import. **c** Straight road with export and import. **d** Straight road. Intersections in SIND[40]. **e** Intersection in Changchun (43.88°N, terrain of low hills with a population of 9.05 million). **f** Intersection in Tianjin (39.08°N, low-lying terrain of coastal plains with a population of 13.63 million). **g** Intersection in Xi'an (34.15°N, terrain of plains and hills with a population of 12.99 million). **h** Intersection in Chongqing (29.35°N, mountainous and hilly terrain with a population of 32.13 million).

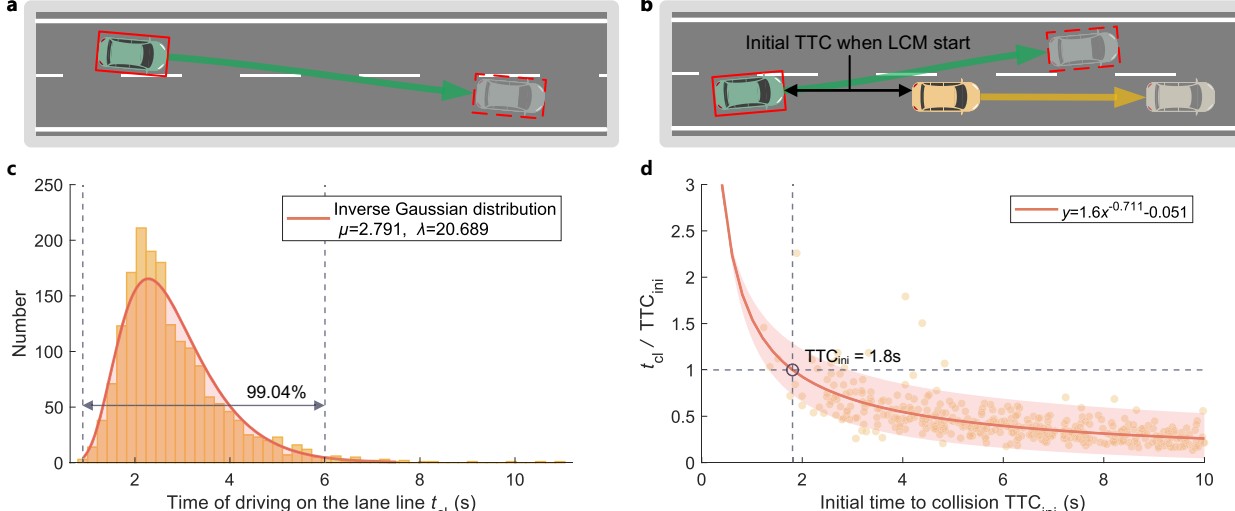

**Fig. 4 | Threshold analysis of the maximum allowable time of driving on the lane line $t_{cl\_max}$ and the minimum allowable TTC with the preceding vehicle $TTC_{cl\_min}$. a** Trigger condition activation period. The trigger is active when the ego vehicle's bounding box overlaps with the lane lines. **b** $TTC_{ini}$ with the preceding vehicle. Once the trigger is active, the TTC with the preceding vehicle is defined as $TTC_{ini}$. **c** Distribution of the time of driving on the lane line. **d** The ratio of $t_{cl}$ to $TTC_{ini}$ in different $TTC_{ini}$ data points and fitting curve. Source data are provided as a Source Data file.

The minimum allowable TTC with the preceding vehicle $TTC_{cl\_min}$ was determined utilizing the AD4CHE dataset to resolve the not impede issue with preceding vehicles using the "No Crashes" principle. A total of 1015 instances out of 1753 complete trajectory data were utilized that the preceding vehicle was slower than the ego vehicle when the ego vehicle made a lane-change. The trigger condition for initiating monitoring is that the bounding box of the ego vehicle overlaps the lane lines and there is a preceding vehicle, the TTC between the ego vehicle and the preceding vehicle is defined as the initial TTC ($TTC_{ini}$). Considering that during the LCM period, $TTC_{ini}$ should not be less than its time of driving on the lane line ($t_{cl}$), there will be a high risk of collisions that impede the preceding vehicle without any additional significant actions (Fig. 4b). $TTC_{ini}$ and $t_{cl}$ were calculated, and their ratio was plotted on the coordinate axis, as shown in Fig. 4d, and fitted. The fitting curve represents the relationship between $TTC_{ini}$ and $t_{cl}$ that most drivers follow during the lane-change period. The $TTC_{ini}$ at the intersection point of the fitted curve and the limiting value is 1.8 s, indicating that most drivers maintained a $TTC_{cl\_min}$ of 1.8 s with the preceding vehicle during the LCM. This is more conservative than the value of 2 s in ECE R157 results, given that ECE R157 considers a deviation of 0.375 m from the lane center line as

the starting point, whereas we utilize the moment of overlapping with the lane line as our starting point, and only a few vehicles can cross that gap by 0.2 s.

The minimum allowable distance from the Rear Vehicle in the Target Lane (RVTL) $d_{cl\_min}$ is determined utilizing the AD4CHE dataset to solve the not impede issue with the RVTL using the "No Crashes" and "No Changes" principles. Two indicators were established to exemplify the "No Changes" principle: the RVTL's minimum acceleration and its deviation from the centerline. These indicators are utilized to evaluate interference by the ego vehicle in both the longitudinal and lateral aspects of the RVTL's behavior. Drawing from the regulation UN ECE R13[44], when the deceleration is less than 0.7 m/s², the braking signal should be suppressed. We adopt this threshold to discern instances of significant braking by the RVTL. This criterion is in accordance with the distribution of the minimum acceleration observed in the RVTL during the LCM of the ego vehicle, as shown in Fig. 5a. Regarding lateral maneuvers, the lateral displacements of lane-keeping vehicles from the lane centerline during the driving process were counted, as shown in Fig. 5b. The 2$\sigma$ interval ([−0.58, 0.37] m) is used as the acceptable wandering zone (WZ) for lane-keeping vehicles, any departure of the RVTL from this defined range during the LCM period of the ego vehicle

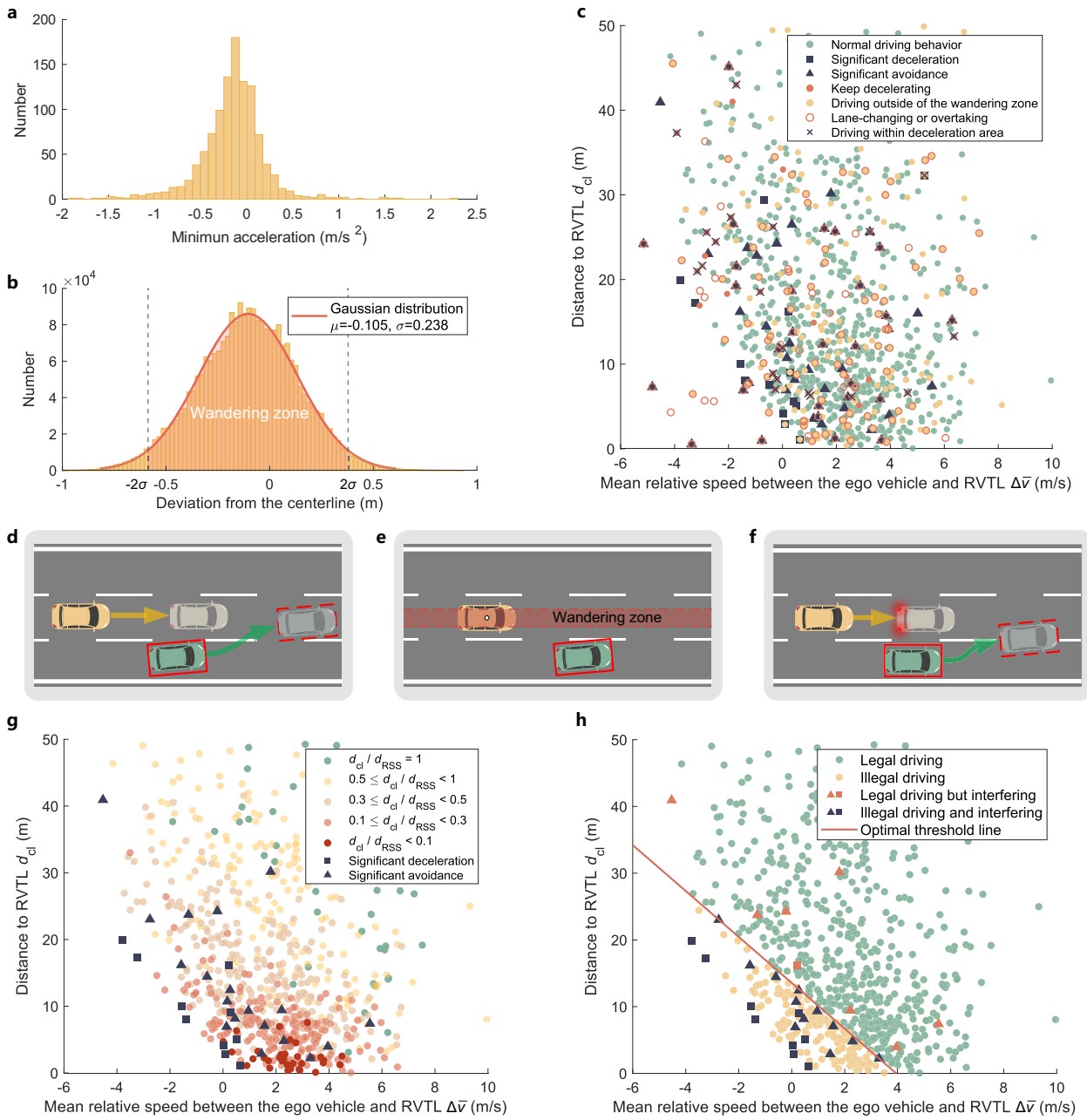

**Fig. 5 | Threshold analysis of the minimum allowable distance from the rear vehicle in the target lane $d_{cl\_min}$. a** Distribution of the minimum acceleration of the RVTL during the LCM. **b** Lateral displacement of lane-keeping vehicles. **c** Different RVTL driving behaviors and their $d_{cl} - \Delta\bar{v}$ data points when the trigger is first activated. **d** Instances retained when the RVTL maintains the target lane throughout the LCM progress. **e** Instances are retained where the RVTL is within the wandering zone when the trigger is first activated. **f** Instances are retained where

RVTL deceleration is less than 0.7 m/s² when the trigger is first activated. **g** RVTL' $d_{cl} - \Delta\bar{v}$ data points marked with different interfered RVTL and $d_{cl}/d_{RSS}$ value levels of the normal drive RVTL when the trigger is first activated in selected instances. **h** Optimal threshold line and RVTL' $d_{cl} - \Delta\bar{v}$ data points marked with different compliance states when the trigger is first activated in selected instances. Source data are provided as a Source Data file.

qualifies as interference. RSS distance was utilized as an indicator to exemplify the "No Crashes" principle. The longitudinal and lateral RSS distances were calculated. The ratios of the current longitudinal and lateral distances between the ego vehicle and RVTL to their corresponding RSS distances were utilized to determine whether the ego vehicle and RVTL maintained a sufficient safety distance.

The corresponding trigger condition is activated when the bounding box of the ego vehicle overlaps the lane line. Once triggered, the ego vehicle and RVTL states were obtained from the dataset. This was calculated, and Fig. 5c shows the distance $d_{cl}$ between the ego vehicle and RVTL when the trigger is activated and the mean relative

speed $\Delta\bar{v}$ between the ego vehicle and RVTL during the LCM of the ego vehicle. The figure also includes color-coded markers of RVTL behavior during the ego vehicle's LCM. RVTLs exhibit various complex behaviors and situations. For example, some RVTLs start to decelerate, while others decelerate continuously; some start to run exit of the wandering zone, while others remain outside the wandering zone or deviate from the centerline within the wandering zone; some attempt to overtake the ego vehicle, while others make lane-change; and some are not on the main road but in a deceleration lane or a deceleration section. These complex behaviors significantly impact the threshold analysis. Consequently, the data are filtered to retain instances with

complete and reasonable behavior to analyze the rational compliance threshold, according to the following criteria: 1) the ego vehicle performs a successful LCM, and 2) the RVTL maintains a constant vehicle ID throughout the scenario (Fig. 5d), 3) the RVTL is within the wandering zone when the LCM starts (Fig. 5e), 4) the RVTL deceleration is less than 0.7 m/s² when the LCM starts (Fig. 5f). Using these criteria, 767 instances were preserved, and 33 RVTLs were clearly interfered with, as shown in Fig. 5g, detailed calculation can be found in Method. Most instances were safe with regard to lateral distances but did not satisfy the longitudinal RSS requirements. The ratios between $d_{cl}$ and the corresponding longitudinal RSS distances $d_{RSS}$ are marked in different colors. Owing to the differences in driver aggressiveness levels, many instances without clear interference were found within the densely populated regions of the interfered instances. Additionally, when driving on highways, vehicles generally do not brake sharply; therefore, most vehicles maintain a small distance that cannot satisfy the RSS distance requirements. However, in the lower-left corner of Fig. 5g, there are no normal driving or non-interfered data points below a certain straight line; thus, people also follow a certain time relationship when making lane-change. Accordingly, different common types of fitting functions are utilized to fit the data, while the linear function performs the best and is defined as a threshold line type, according to sensitivity analysis and Supplementary Result. The final filtered data were optimized using a genetic algorithm with the optimization objective set to minimize the false-negative rate while ensuring a false-positive rate. The resulting threshold line is shown in Fig. 5h. Finally, when monitoring, the $d_{cl\_min}$ value was determined as follows:

$$d_{cl\_min} = \begin{cases} 50, & \Delta v < -10.7 \\ -3.4\Delta v + 13.6, & -10.7 \le \Delta v \le 4 \\ 0, & \Delta v > 4 \end{cases} \quad (1)$$

where $\Delta v$ represents the speed difference in m/s between the ego vehicle and RVTL at each sample time when the trigger is activated.

The minimum allowable time difference to intersection point $TTI_{diff\_min}$ is determined using the SIND dataset, and it addresses whether left-turning vehicles do not interfere with straight-moving vehicles at an intersection with shared traffic lights for left-turning and straight-moving vehicles. Using the SIND dataset, data instances were retained in which the time interval between left-turning and straight-moving vehicles passing through the intersection point of their trajectories was less than 5 s, resulting in 554 recorded instances. Utilizing the filtered instances, the positions of the intersection points of the left-turning and straight-moving vehicle trajectories were obtained. The time sequences $TTI_{left}$ and $TTI_{str}$ required for the left-turning vehicle and straight-moving vehicles to reach their corresponding intersection point were calculated according to the speed and distance to the intersection point in each frame. The $TTI_{left}$ and $TTI_{str}$ for each instance are presented in Fig. 6a. Circular marker points correspond to the data points in Fig. 6b. As shown, in scenarios involving conflicts between left-turning and straight-moving traffic, two main typical situations exist: 1) Straight-moving vehicles pass the conflict zone first (including left-turn vehicles give way). 2) Left-turn vehicles pass through the conflict zone first (including the left-turn vehicle rush). To assess compliance with intersection regulations, it is customary to establish virtual lane boundaries to analyze vehicle movements. For a typical unprotected left-turn intersection, an example conflict scenario between left turning and straight movement is shown, and the corresponding virtual lane lines are presented in Fig. 6c.

The trigger condition was active when the bounding box of the left-turn vehicle overlapped with the conflict area. When the trigger starts to activate, according to the principle of "No Crashes", the adaptive RSS distance $d_{aRSS}$ of the straight-moving vehicle and the current distance $d_{realstr}$ from the straight-moving vehicle to the

intersection point are calculated, as well as the $TTI_{left,act}$ and $TTI_{str,act}$ at the trigger activation time. All the data from the filtered conflict scenarios are shown in Fig. 6b, where $TTI_{diff} = TTI_{str,act} - TTI_{left,act}$. $d_{aRSS}$ is utilized to determine whether a straight-moving vehicle has a sufficiently safe distance when the trigger is activated. If $d_{realstr} - d_{aRSS}$ is less than zero, a straight-moving vehicle is regarded as unsafe. According to the principle of "No Changes", the minimum acceleration $a_{min}$ of the straight-moving vehicle is calculated when the trigger is activated until the left-turning or straight-moving vehicle reaches the intersection point, as shown in Fig. 6d. $a_{min}$ is utilized to determine whether the left-turning vehicle interferes with the straight-moving vehicle. In accordance with the ECE R13 guidelines, if $a_{min}$ is less than -0.7 m/s², the straight-moving vehicle is regarded as being interfered. When left-turning vehicles cross the conflict area ahead of the others, there is a potential for violations—particularly when left-turning vehicles rush through. Therefore, a threshold analysis based on the data where left-turning vehicles take precedence was conducted, and 341 instances remained. Utilizing the data of the 341 instances when the trigger was activated, the difference time $TTI_{diff}$ was calculated. Utilizing $TTI_{diff}$ as the horizontal axis and the difference between $d_{aRSS}$ and $d_{realstr}$ as the vertical axis, Fig. 6e was created. The data points are color-coded to indicate the safety or interference of straight-moving vehicles. This clearly illustrates that data points that cannot meet the "No Crashes" and "No Changes" principles mainly fall within the $TTI_{diff}$ interval of 3–5 s. Some drivers exhibit aggressive driving behavior, even when $TTI_{diff}$ is small, implying that they do not take precautions. In contrast, some drivers adopted a relatively conservative approach, making noticeable avoidance maneuvers even when $TTI_{diff}$ was relatively large. Through sensitivity analysis, the $TTI_{diff\_min}$ was selected as 3.4 s. It is worth noting that, the slope of threshold line of $d_{cl\_min}$ indicates a time relationship between the ego vehicle and RVTL is also 3.4s. This is not a coincidence. It shows that regardless of the road structure and the scene, the acceptable time difference for mutual impede and interference is consistent.

## Sensitivity analysis

A sensitivity analysis was performed on key thresholds such as $d_{cl\_min}$ and $TTI_{diff\_min}$. For $d_{cl\_min}$, our goal is to find an optimal function that maximizes the weighted sum of the true positive (non-compliance and judged as non-compliance) number and true negative (compliance and judged as compliance) number, while ensuring a certain false-positive rate, as explained in Method Eqs. (21) and (22). The optimal cost and corresponding false-positive rate and false-negative rate results for different maximum allowable false-positive rates (0% to 30%) are shown in Fig. 7b, and part of the optimal threshold line $d_{cl\_min}$ are shown in Fig. 7a. As the data points are closer to the lower-left corner, the corresponding lane-change maneuvers pose a higher risk. Consequently, to reduce false-negative outcomes under different false-positive tolerances, the threshold line was gradually shifted toward the upper-right corner. It is evident that as the false-positive requirements become more relaxed, the false-negative rate decreases, and the result exhibits a staircase pattern. The maximum cost is 1010.7 with a 24% false-positive rate and the numbers of false-positive and false-negative are 160 and 8. In the next step platform, the false-positive rate will become greater than the false-negative rate, which is not what we expect. Consequently, the $d_{cl\_min}$ threshold line was selected according to a 24% false-positive rate, as shown in Eq. (1).

For $TTI_{diff\_min}$, as the function form is fixed, there is no need for algorithmic optimization. Therefore, We statistics the weighted sum values (Method Eq. (22).) and corresponding false-negative rates at different false-positive rates, as shown in Fig. 7d. Under the different false-positive rates (0% to 20%), the thresholds are illustrated in Fig. 7c. The maximum cost is 297.8 with a 6% false-positive rate and the numbers of false-positive and false-negative are 9 and 42. When the false-positive rate exceeds 18%, the false-positive rate will become

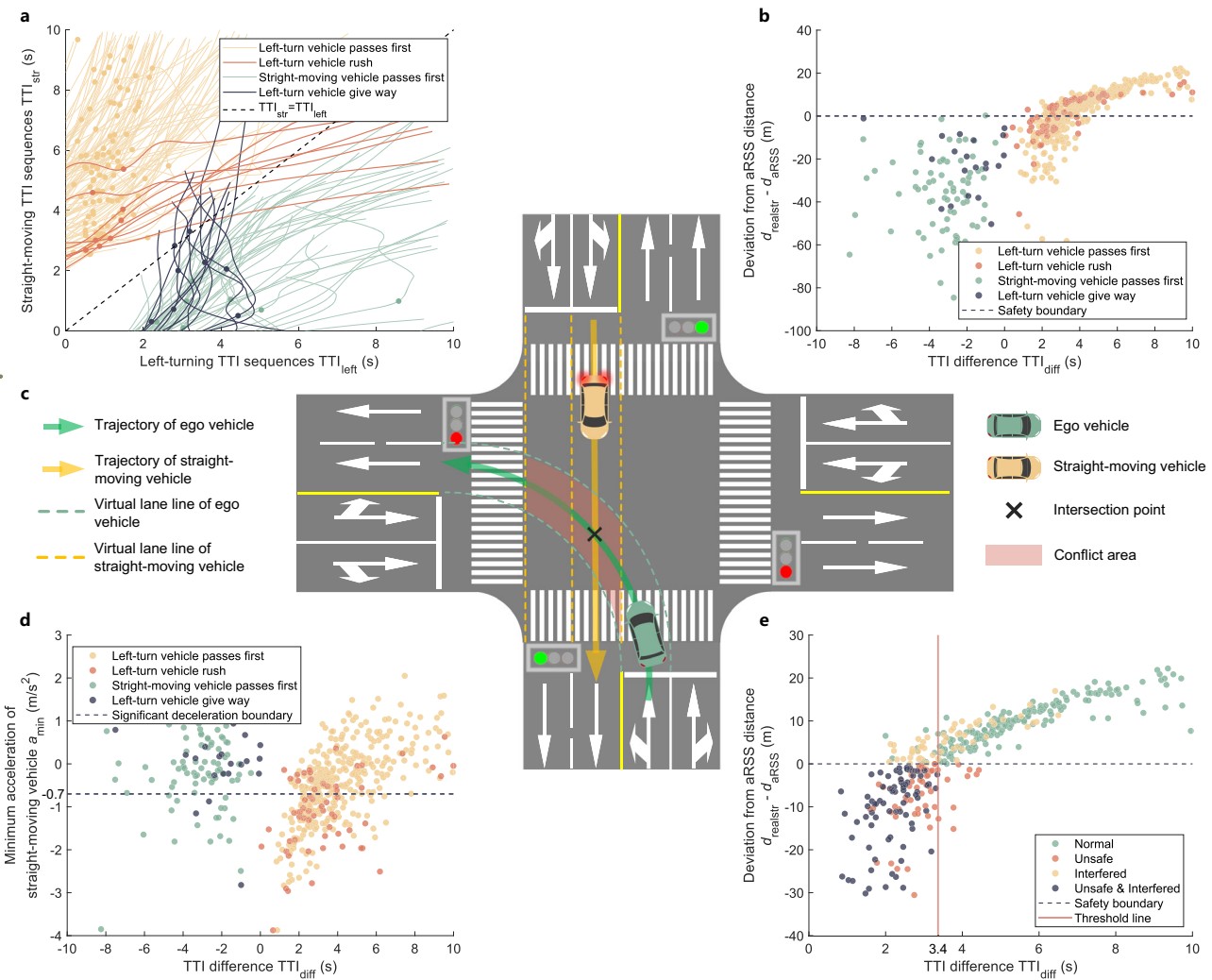

**Fig. 6 | Threshold analysis of the minimum allowable time difference between left-turn vehicle and straight-moving vehicle to intersection point TTI$_{diff\_min}$.** **a** Times to the intersection point of left-turn and straight-moving vehicles. As the vehicle moves towards the intersection point, the data points also move along the curve towards the horizontal or vertical axis. The marked data points on each line correspond to the data points in Fig. 6b. **b** Safety distance indicator of straight-moving vehicles at the time of activation of the trigger condition. Points below 0 m are considered unsafe. **c** Virtual lane lines and conflict areas of a left-turn and straight-moving conflict scenario. **d** Interference indicator of straight-moving vehicles at the time of activation of the trigger condition. The points below −0.7 m/s² are regarded as interfered. **e** States of straight-moving vehicles when left-turning vehicles pass the conflict zone first. Source data are provided as a Source Data file.

greater than the false-negative rate. Therefore, TTI$_{diff\_min}$ of 3.4 s with a false-positive rate of 6% was selected as the threshold.

## Monitoring results on datasets

According to the thresholds and MTL expression, an online monitor was utilized for every vehicle in the dataset. The selected vehicle was treated as the ego vehicle, whereas the other vehicles in the scenario were treated as surrounding vehicles. All the necessary information was transmitted to the online monitor by the designed data bus in a dataset sampling time sequence that contained ego vehicle parameters, traffic signs, traffic participants, and map data. According to the selected traffic-law articles, six types of violation behaviors were monitored. The statistical results are shown in Table 1. Figure 8a presents the count of activated trigger conditions for each monitor type in a bar chart. The number of vehicles that violate the corresponding article during trigger condition activation is color-coded and accompanied by a corresponding percentage.

For the AD4CHE dataset, in vehicles with corresponding trigger condition activated, there are a total of 18017 vehicles counting for 95.06% violate the speed limitation, a total of 15423 vehicles counting

for 84.46% violate the following distance limitation, a total of 169 vehicles counting for 4.04% violate the drive on lane line limitation, and a total of 718 vehicles counting for 19.25% violate the lane-change limitation. Because of the slight congestion on the road, it is difficult for most vehicles to satisfy the minimum speed and distance requirements, resulting in a large proportion of violations. When making lane-change and overtaking, fewer vehicles violate the laws. Approximately 4.04% of vehicles drive on lane lines for over 6 s, which may be caused by a curved road, making it difficult to drive in the lane. Out of 718 instances of lane-change violations, there are 704 vehicles that keep an insufficient distance with RVTL because aggressive drivers cannot maintain a sufficient safe distance from the RVTL when making lane-change.

Statistical violation results for the 25th fragment in the AD4CHE dataset and typical illegal examples are shown in Fig. 8b. This fragment lasted 290 s and contains 786 trajectories. The statistics for each type of violation were counted at intervals of 5 s. Among them, vehicle 9629 ran in the second inner lane at a speed of 55.5 km/h (far lower than the minimum speed limitation of 100 km/h) with a following distance of 19.7 m (far shorter than the minimum compliance following distance of

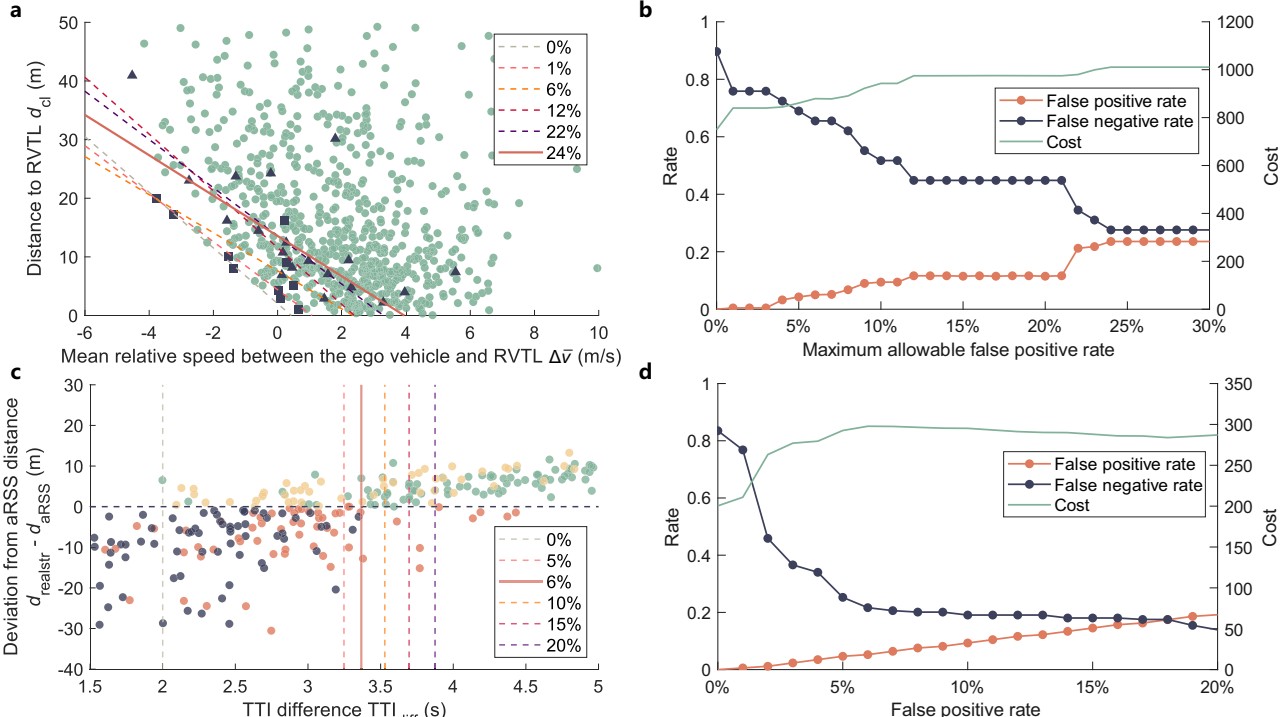

**Fig. 7 | Sensitivity analysis of the minimum allowable distance from the rear vehicle in the target lane $d_{cl\_min}$ and the minimum allowable time difference between the left-turning vehicle and straight-moving vehicle to intersection point $TTI_{diff\_min}$. a** Threshold lines under different maximum allowable false-positive rates. **b** False-negative rate and the cost under different maximum allowable false-positive rates. **c** Thresholds under different false-positive rates. **d** False-negative rates and the cost under different false-positive rates. Source data are provided as a Source Data file.

**Table 1 | The counts of vehicle violations on datasets**

| Traffic law violation behaviors | | Number of traffic law violation instances | Percentage of traffic law violation instances | Number of total monitoring instances |
|---|---|---|---|---|
| Speed violation | | 18017 | 95.06% | 18,953 |
| Following distance violation | | 15,423 | 84.46% | 18,261 |
| Driving on lane line | | 169 | 4.04% | 4181 |
| Lane-change violation | Insufficient TTC with front vehicle& insufficient distance with RVTL | 7 | 0.19% | 3729 |
| | Insufficient TTC with front vehicle | 7 | 0.19% | 3729 |
| | Insufficient distance with RVTL | 704 | 18.88% | 3729 |
| Traffic light violation | Run the yellow light | 218 | 1.68% | 12,994 |
| | Run the red light | 545 | 4.19% | 12,994 |
| | On stop line at the yellow light | 54 | 0.42% | 12,994 |
| | On stop line at the red light | 471 | 3.62% | 12,994 |
| Right-of-way violation | | 101 | 1.85% | 5467 |

50 m). Vehicle 9929 violated the lane-change article because of its short distance from the RVTL according to the compliance threshold in Eq. (1). Vehicle 10053 drove on the lane line for approximately 7.4 s, which exceeded the compliance threshold of 6 s.

For the SIND dataset, in vehicles with the corresponding trigger condition activated, there are a total of 1288 vehicles counting for 9.91% violating the traffic light limitation, and a total of 101 vehicles counting for 1.85% violate the right of way limitation. More detailed statistics on different types of violations related to traffic lights in different regions can be found in Supplementary Result. The statistical violation results for the 8_2_1 fragment in the SIND dataset are presented in Fig. 8c. This fragment lasted 1200 s and contained 611 trajectories. The statistics of the intersection violations for each type at intervals of 20 s are presented. Among them, vehicle 78 violated the

right-of-way with a $TTI_{diff}$ of 3.2 s, interfering with the straight movement of vehicle 73. Vehicle 365 violated the traffic light law, it ran the yellow light after the yellow light turned on 0.93 s ago. Vehicle 481 violated the traffic light laws. It crossed the stop line when the red light was on.

### Field testing

To demonstrate the rationality and real-time capabilities of the proposed online legal driving behavior monitor, the monitoring system was developed using C++ and integrated into an IPC installed in a FAW Jiefang commercial SV (Fig. 9a). A data-transmission bus was established to satisfy the specific data requirements. Real-time information exchanges between the autonomous driving and online monitoring systems via in-vehicle Ethernet, adhering to the data bus format. This

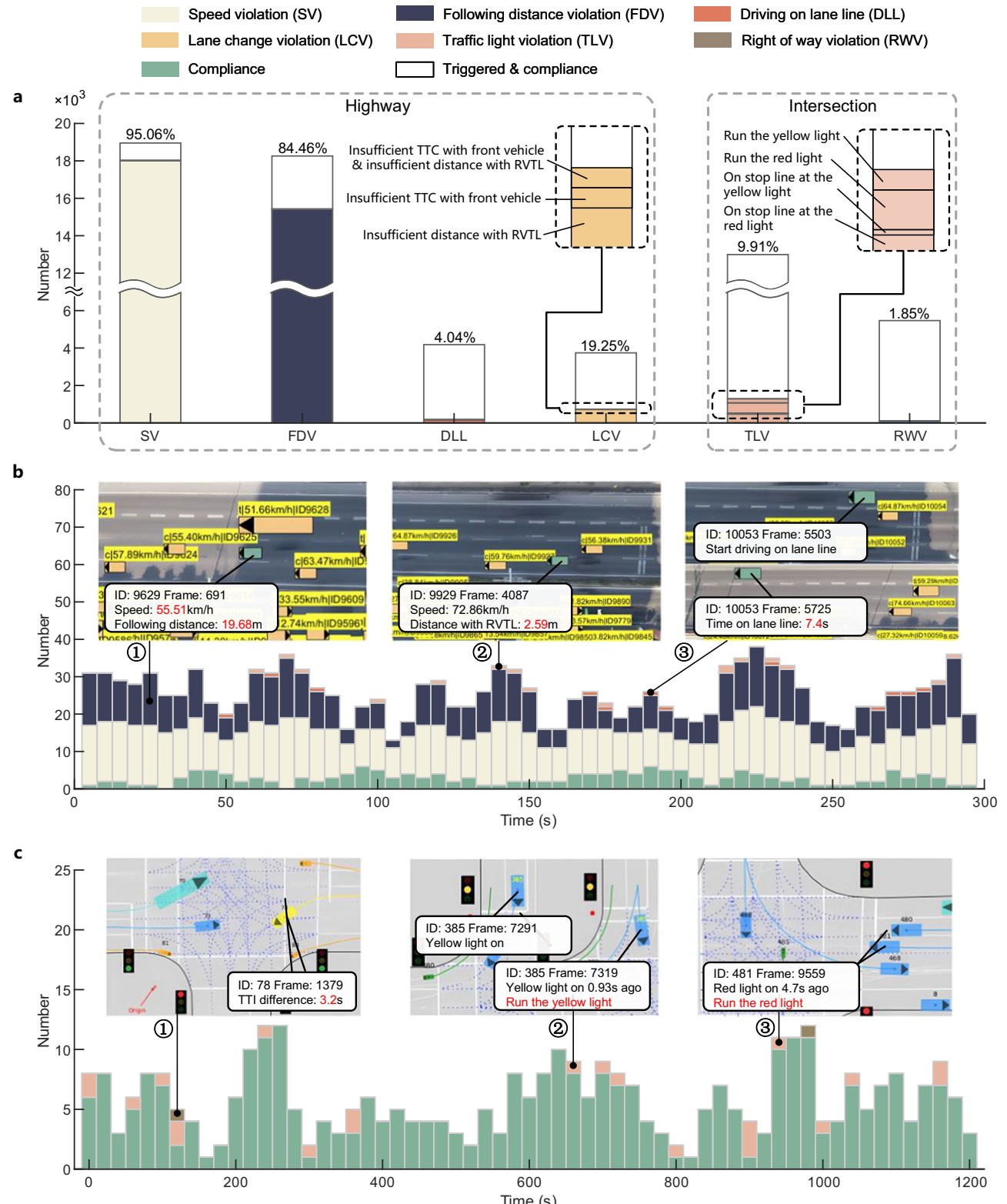

**Fig. 8 | Results of dataset validation. a** Statistical online monitoring results of dataset validation. **b** Statistical online monitoring results and typical illegal examples for the 25th fragment in the AD4CHE dataset. **c** Statistical online monitoring results and typical illegal examples for the 8_2_1 fragment in the SIND dataset. Source data are provided as a Source Data file.

exchange included the vehicle configuration, status of the ego vehicle and surrounding objects, map data, and trajectory of the ego vehicle. Real-vehicle tests were conducted at the Intelligent Connected Vehicle (ICV) Testing Base (Fig. 9b). Three typical traffic-law violation scenarios were considered to verify the effectiveness and real-time capabilities

of the online monitoring system. The replay video can be found in the Supplementary Movie 1– 4.

Distance limitation violation: The target vehicle (TV) occupied the same lane as the ego vehicle and was placed 60 m ahead. Both vehicles began to move simultaneously. The TV accelerated to 30 km/h and

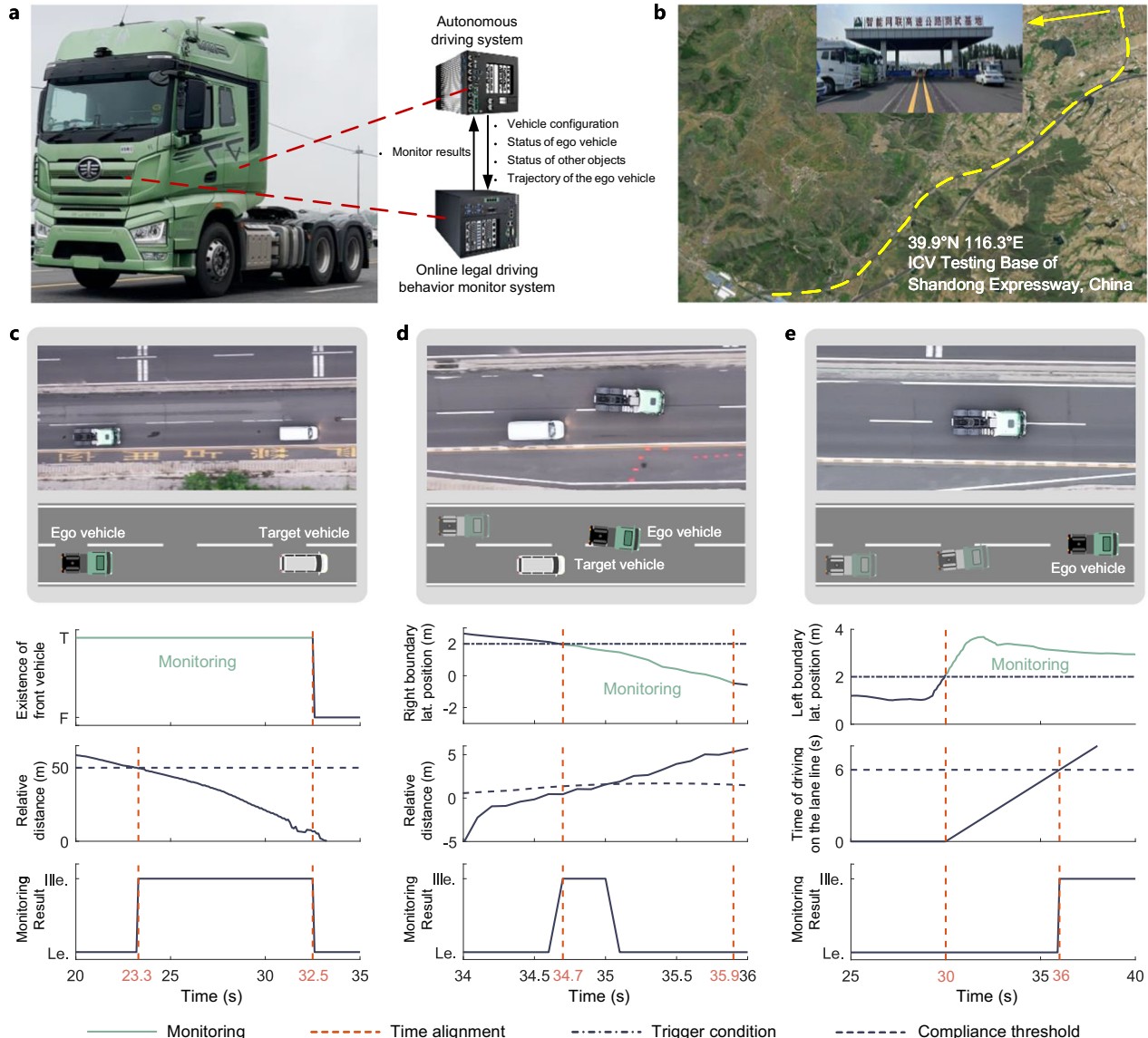

**Fig. 9 | Field testing and results. a** Test commercial vehicle with the online legal driving behavior monitor. **b** Test section at Intelligent Connected Vehicle Testing Base of Shandong Expressway in China. **c** Images, trigger time, and monitoring result of distance limitation violation. **d** Images, trigger time, and monitoring result of lane-change violation. **e** Images, trigger time, and monitoring result of driving on lane line violation. Source data are provided as a Source Data file.

maintained this speed, whereas the target speed of the SV was 60 km/h, as shown in Fig. 9c. The trigger condition for distance limitation violation monitoring was activated when the vehicle was on a highway and a preceding vehicle was detected in the same lane. The following distance was reduced to 50 m at 23.3 s, and the monitor accurately identified the SV violating the regulation to maintain a safe following distance. The violation persisted until 32.5 s when the SV made a left lane-change and there was no longer a preceding vehicle in the same lane.

Lane-change violation: The SV performed a lane-change maneuver by cutting in front of the TV, as shown in Fig. 9d. The trigger condition for illegal lane-change monitoring was activated when the SV overlapped the lane line and had a lateral velocity. A RVTL was detected and the relative distance and velocity between the SV and RVTL were calculated. The calculation results fell below the compliance threshold, indicating an illegal lane-change violation at 34.7 s.

Driving on lane line violation: The SV initially traveled along the centerline of the lane but then deviated leftward, causing its left boundary to cross into the left lane; thus, it drove on the lane line, as shown in Fig. 9e. The trigger condition for long-term driving on lane line monitoring was activated when the SV overlapped the lane line at 30 s. The SV continued to drive along the lane line. 6 s later (at 36 s), the monitor detected a violation of long-term driving on the lane line.

## Discussion

In this work, a trigger-based hierarchical online monitor was proposed, along with driving behavior-based ambiguous compliance thresholds determination, to ensure the rational and fact-based legal judgments align with common understanding. This work has three notable advantages. First, a trigger-based hierarchical structure for the online monitor only triggers the articles that should be monitored and utilizes the driving behavior to increase the rationality of monitoring. Second, ambiguous human-oriented traffic laws can be implemented into this online monitor via compliance threshold determination utilizing general principles. Finally, the monitor results based on the drone datasets and field test demonstrate the feasibility and rationality of the proposed online legal driving behavior monitor.

However, the process of establishing precise, specific compliance thresholds for evaluating complex behaviors of traffic participants requires further data validation. Constructing and updating joint distribution parameters[45,46] for critical behavioral variables and deriving specific probabilities of behavioral violations from a larger dataset of real-world violations may yield more robust results. Equally important is the need for feedback mechanisms involving government agencies and various stakeholders to collaboratively define the ultimate compliance threshold for widespread implementation in SVs.

It is essential to acknowledge that traditional traffic laws, designed for human drivers, may not fully satisfy the requirements of SVs in the future. Leveraging the quantitative data processing capabilities of SVs, we have the opportunity to develop more detailed traffic regulations tailored specifically to them. These regulations can strike a balance between safety, efficiency, and compliance in complex scenarios while taking into account human factors and ethical considerations, thereby promoting the legal implementation of SVs.

## Methods

### Metric temporal logic

Temporal logic has been widely utilized in studies on traffic law formalization. Because the proposed monitor contains continuous state judgments, we utilized MTL to interpret the definitions of the trigger conditions and logic judgments. MTL introduces more temporal operators and performs well in describing the relationship between behavioral logic within limited temporal and spatial coordinates.

Given a set AP of atomic propositions, where each atomic proposition $\sigma_i \in$ AP represents a Boolean statement, the MTL formula $\varphi$ is defined as follows:

$$\varphi ::= \quad \sigma_i | \neg\varphi | \varphi_1 \wedge \varphi_2 | \varphi_1 \vee \varphi_2 | \varphi_1 \Longleftrightarrow \varphi_2 | \varphi_1 \Rightarrow \varphi_2 | \\ G_I(\varphi) | F_I(\varphi) | P_I(\varphi) | O_I(\varphi) | \varphi_1 U \varphi_2 \tag{2}$$

where $\neg$, $\wedge$, and $\vee$ are the Boolean operators, $\varphi_1 \Leftrightarrow \varphi_2$ indicates that $\varphi_1$ is equivalent to $\varphi_2$, and $\varphi_1 \Rightarrow \varphi_2$ indicates that when $\varphi_1$ is satisfied, it is necessary that $\varphi_2$ is satisfied. $G$, $F$, $P$, and $O$ are temporal operators. The subscript $I$ represents the interval $\mathbb{R}_{\geq 0}$, which expresses the time constraints relative to the current time. The global operator $G$ indicates that $\varphi$ holds throughout the entire time sequence, and the future operator $F$ indicates that $\varphi$ holds within a time interval for some future states. The previous operator $P$ states that $\varphi$ holds within a time interval for the previous state, and the once-operator $O$ specifies that $\varphi$ holds at least once before a certain point in time. The until operator $U$ specifies that a property holds true until another property becomes true.

### Trigger-based hierarchical online monitor architecture

We established a standardized process for formalizing traffic laws, and the detailed steps are presented in detail in the Supplementary Method. This process facilitates the conversion of each regulator into a corresponding MTL. When systematically considering all articles, it is important to recognize that different articles may share common elements. Some articles may even include simpler ones, as depicted in the lower-left corner of Fig. 1. Complete overtaking behavior comprises two lane-changing processes, each involving a driving segment on the lane line. Therefore, we designed the monitoring system with a trigger-based hierarchical monitoring architecture. Each MTL expression of an article is represented as Trigger condition $\Rightarrow$ Logical judgment, where the sequence of atomic propositions in Trigger condition should be strictly adhered to, as it signifies the hierarchical relationship

among various triggering conditions:

$$\text{Trigger condition} \Leftrightarrow \underbrace{\underbrace{\underbrace{T_1 \wedge T_2}_{S1} \wedge ... \wedge T_n}_{S2}}_{Sn} \tag{3}$$

Consider a set $S2 = \{T_1, T_2\}$ as the trigger set of article $A2$, where trigger set $S1 \subseteq S2$ represents the trigger set of article $A1$. Then, $A2$ is the parent article of $A1$. When the monitoring system receives the input requirements information, it evaluates the triggers in the bottom-level articles. If the specified conditions are satisfied, the article is monitored, and the input information is transmitted to its parent article for evaluation of any remaining trigger subsets. This comprehensive trigger-based hierarchical relationship can be expressed as $((T_1 \Rightarrow L_1) \wedge T_2 \Rightarrow L_2) \wedge ... \wedge; T_n \Rightarrow L_n$, as depicted at the bottom of Fig. 1. This systematic approach ensures that each layer is progressively traversed. Therefore, real-time monitoring is necessary only for articles relevant to the current situation and scenario, to optimize computational resource utilization. Table 2 presents the trigger conditions and logical judgments of the articles considered in this study.

Many regulations involve the attributes of road networks that are difficult to obtain solely through vehicle perception. Considering the need for SV deployment, we obtain road network attribute information through either maps or vehicle perception systems if the required information can be provided. For highways, each lane has an ID L and road type RT. We have RT = {M, R, A, D, E}, which represents the mainline, ramp, acceleration lane, deceleration lane and emergency lane, respectively. L of the innermost lane in the same direction is 1, and the IDs of the other lanes increase outward. Each lane's left lane line ID is consistent with the lane ID. The lane line with ID $i$ is expressed as y($i$) and is represented by a cubic fitting curve. In addition, the content information of traffic signs must be provided.

The information of the other traffic participants is represented in the vehicle coordinate system. The coordinate origin is located at the center of the bounding box. The state x = [$X$, $Y$, $\theta$, vx, vy] of a vehicle consists of the longitudinal coordinates $X$, lateral coordinates $Y$, heading angles $\theta$, longitudinal velocities vx and lateral velocities vy. The operator x(obj) denotes the value of the state $x$ for participant obj. According to the geometrical parameters and heading angle $\theta$ of obj, the planar area occupied by obj, which is denoted as Area(obj), can be calculated according to the geometric parameters and heading angle $\theta$ of obj. Using lane line data, the area around the ego vehicle can be divided into six regions: Front(Ego), FrontLeft(Ego), FrontRight(Ego), Rear(Ego), RearLeft(Ego), and RearRight(Ego). The closest TV Tgt in each region is defined as Tgt$_{(\cdot)}$, where the subscript $(\cdot)$ represents the

**Table 2 | Trigger condition and logical judgments of typical articles**

| Article | Trigger condition | Logical judgment |
|---|---|---|
| Article78 | $T_{RT} \wedge T_{Spd1}$ | vx(Ego) $\in [V_{sign\_min}, V_{sign\_max}]$ |
| | $T_{RT} \wedge \neg T_{Spd1} \wedge \neg T_{Spd2} \wedge T_{Spd3}$ | vx(Ego) $\in [100, 120]$ |
| Article80 | $T_{RT} \wedge T_{Dis1} \wedge \neg T_{Dis2}$ | dis(Ego, Tgt$_f$) > 50 |
| Article82.6 | $T_{onRLine}$ | $\neg(t_{now} - t_{in} > t_{cl\_max})$ |
| | $T_{onLLine}$ | $\neg(t_{now} - t_{in} > t_{cl\_max})$ |
| Article44 | $T_{onLLine} \wedge T_{pvy}$ | $\neg$(FViolation $\vee$ RLViolation) |
| Article38.1 | $T_{TL1}$ | $\neg$ TL(Ego) = R |
| | $T_{TL1} \wedge T_{TL2}$ | $P_{[t_{in}, t_{now}]}(T_{TL1}) \wedge P_{[t_y, t_{now}]}(T_{TL2}) \wedge t_y > t_{in}$ |
| Article38.2 | $T_{Intersection} \wedge T_{ls}$ | $\neg$ RWViolation_LS |

corresponding region; for example, $\text{Tgt}_{fl}$ represents the vehicle closest to the ego vehicle in the FrontLeft(Ego) region. Furthermore, certain concepts are utilized to calculate the Boolean values of the atomic propositions including the following: $L(\text{obj}, t)$ represents the lane ID to which obj belongs at time $t$. $\text{dis}(\text{obj}_1, \text{obj}_2)$ and $\text{TTCX}(\text{obj}_1, \text{obj}_2)$ represent the longitudinal distance and time to collision between $\text{obj}_1$ and $\text{obj}_2$ along the lane direction, respectively. $\text{overlap}(r_1, r_2)$ is utilized to determine whether there is an overlap between the regions $r_1$ and $r_2$. According to the calculation of the position of obj and speed signs, $\text{SpdSignArea}(\text{obj})$ represents the obj located in the speed-limit sign management area. More information about the concepts involved in atomic propositions can be found in Supplementary Information.

Article 78 requires monitoring while driving along a highway mainline. The speed limitation can be affected by factors such as the number of lanes and the lane in which the vehicle is positioned. Thus, the trigger set for Article 78 is defined as follows:

$$\begin{aligned} T_{\text{RT}} &\Longleftrightarrow \text{RT}(\text{Ego}) = M \\ T_{\text{Spd1}} &\Longleftrightarrow \text{SpdSignArea}(\text{Ego}) \\ T_{\text{Spd2}} &\Longleftrightarrow N_{\text{ml}} \geq 3 \\ T_{\text{Spd3}} &\Longleftrightarrow L(\text{Ego}, t_{\text{now}}) = 1 \end{aligned} \quad (4)$$

where $N_{\text{ml}}$ represents the number of mainlines in the same direction and $t_{\text{now}}$ represents the current time.

In the monitoring system, the speed monitor corresponding to Article 78 is expressed as follows:

$$\begin{aligned} T_{\text{RT}} \wedge T_{\text{Spd1}} &\Rightarrow \text{vx}(\text{Ego}) \in [V_{\text{sign\_min}}, V_{\text{sign\_max}}]\text{km/h} \\ T_{\text{RT}} \wedge \neg T_{\text{Spd1}} \wedge \neg T_{\text{Spd2}} \wedge T_{\text{Spd3}} &\Rightarrow \text{vx}(\text{Ego}) \in [100, 120]\text{km/h} \end{aligned} \quad (5)$$

where $V_{\text{sign\_max}}$ and $V_{\text{sign\_min}}$ represent the upper and lower limits, respectively, as indicated by the speed-limit sign.

The trigger condition in Article 80 is that the ego vehicle is on the highway mainline and has a preceding vehicle.

$$\begin{aligned} T_{\text{Dis1}} &\Longleftrightarrow \exists \text{Tgt}_f \\ T_{\text{Dis2}} &\Longleftrightarrow \text{vx}(\text{Ego}) \geq 100\text{km/h} \end{aligned} \quad (6)$$

Monitoring of the following distance starts when the relevant triggers are activated. The expression is as follows:

$$T_{\text{RT}} \wedge T_{\text{Dis1}} \wedge \neg T_{\text{Dis2}} \Rightarrow \text{dis}(\text{Ego}, \text{Tgt}_f) > 50\,\text{m} \quad (7)$$

The trigger condition of Article 82.6 is that the ego vehicle overlaps the lane line, which can be expressed as follows:

$$\begin{aligned} T_{\text{onLLine}} &\Longleftrightarrow P_{[t_{\text{in}}, t_{\text{now}}]}(\text{overlap}(\text{Area}(\text{Ego}), y(L(\text{Ego}, t_{\text{in}})))) \\ T_{\text{onRLine}} &\Longleftrightarrow P_{[t_{\text{in}}, t_{\text{now}}]}(\text{overlap}(\text{Area}(\text{Ego}), y(L(\text{Ego}, t_{\text{in}}) + 1))) \end{aligned} \quad (8)$$

The time at which the trigger condition begins to be satisfied is denoted as $t_{\text{in}}$. When the relevant triggers are satisfied, the monitoring of drive over or on the dividing line begins.

$$\begin{aligned} T_{\text{onLLine}} &\Rightarrow \neg(t_{\text{now}} - t_{\text{in}} > t_{\text{cl\_max}}) \\ T_{\text{onRLine}} &\Rightarrow \neg(t_{\text{now}} - t_{\text{in}} > t_{\text{cl\_max}}) \end{aligned} \quad (9)$$

Article 44, illustrates this with the example of a lane-change to the left: the trigger condition is defined as the moment when the ego vehicle overlaps the left-lane line with a lateral speed greater than zero.

$$\begin{aligned} T_{\text{pvy}} &\Longleftrightarrow \text{vy}(\text{Ego}) > 0 \\ T_{\text{nvy}} &\Longleftrightarrow \text{vy}(\text{Ego}) < 0 \end{aligned} \quad (10)$$

During this maneuver, the ego vehicle must maintain a non-interfering distance from the other relevant vehicles, including the

front vehicle and RVTL. The specific expressions are as follows:

$$\text{FViolation} \Longleftrightarrow P_{[(T_{\text{onLLine}} \wedge T_{\text{pvy}})_1]}(\exists \text{Tgt}_f \wedge TTCX(\text{Ego}, \text{Tgt}_f) \leq \text{TTC}_{\text{cl\_min}}) \quad (11)$$

$$\text{RLViolation} \Longleftrightarrow \exists \text{RVTL} \wedge \text{dis}(\text{RVTL}, \text{Ego}) \leq d_{\text{cl\_min}} \quad (12)$$

where $(T_{\text{onLLine}} \wedge T_{\text{pvy}})_1$ represents the first moment within the time interval when $(T_{\text{onLLine}} \vee T_{\text{pvy}})$ is true.

The lane-change process involves a segment of driving on the lane line, and driving on the lane line should be monitored, to ensure that the vehicle does not remain on the lane line for an extended duration during the lane-change process. The corresponding expressions is as follows:

$$T_{\text{onLLine}} \wedge T_{\text{pvy}} \Rightarrow \neg(\text{FViolation} \vee \text{RLViolation}) \quad (13)$$

Intersection areas involve many right-of-way relationships and often lack well-defined lane lines. Therefore, a map is needed to provide virtual lane (Fig. 5c) information. IntersectionArea is defined as the region formed by extending the intersection from the stop line StopLine. The traffic light states $\text{TL} = \{R, G, Y\}$ represent the red, green, and yellow lights, respectively. In addition, $\text{TTIdiff}(\text{obj}_1, \text{obj}_2)$ donates the difference TTI to trajectory intersection point between $\text{obj}_1$ and $\text{obj}_2$.

The trigger condition for Article 38.1 is that the ego vehicle overlaps with StopLine and the traffic light is yellow or red.

$$\begin{aligned} T_{\text{TL1}} &\Longleftrightarrow \text{overlap}(\text{Area}(\text{Ego}), \text{StopLine}) \\ T_{\text{TL2}} &\Longleftrightarrow \text{TL}(\text{Ego}) = Y \end{aligned} \quad (14)$$

The ego vehicle should not enter the intersection under a red light. When the yellow light is on and the ego vehicle has not yet entered the intersection, it should not enter the intersection.

$$\begin{aligned} T_{\text{TL1}} &\Rightarrow \neg \text{TL}(\text{Ego}) = R \\ T_{\text{TL1}} \wedge T_{\text{TL2}} &\Rightarrow P_{[t_{\text{in}}, t_{\text{now}}]}(T_{\text{TL1}}) \wedge P_{[t_{\text{y}}, t_{\text{now}}]}(T_{\text{TL2}}) \wedge t_{\text{y}} > t_{\text{in}} \end{aligned} \quad (15)$$

where $t_{\text{in}}$ represents the moment when the ego vehicle starts to overlap with the stop line and $t_{\text{y}}$ represents the initial moment when the yellow light turns on.

In this study, we analyzed the right-of-way using the example of a left-turn and straight-moving conflict scenario. The corresponding trigger condition is when a left-turning vehicle intrudes into the virtual lane line of the oncoming straight-moving traffic.

$$\begin{aligned} T_{\text{Intersection}} &\Longleftrightarrow \text{overlap}(\text{Area}(\text{Ego}), \text{IntersectionArea}) \\ T_{\text{ls}} &\Longleftrightarrow \text{overlap}(\text{Area}(\text{Ego}), \text{VirtualLane\_O}) \end{aligned} \quad (16)$$

Where VirtualLane_O represents the virtual lane of oncoming straight-moving traffic.

When the ego vehicle intrudes into the virtual lane line of the oncoming straight-moving traffic, it should ensure that TTIdiff exceeds $\text{TTI}_{\text{diff\_min}}$. The specific expressions are as follows:

$$\text{RWViolation\_LS} \Longleftrightarrow \exists \text{Tgt}_{\text{os}} \wedge \text{TTIdiff}(\text{Ego}, \text{Tgt}_{\text{os}}) \leq \text{TTI}_{\text{diff\_min}} \quad (17)$$

$$T_{\text{Intersection}} \wedge T_{\text{ls}} \Rightarrow \neg \text{RWViolation\_LS} \quad (18)$$

where $\text{Tgt}_{\text{os}}$ represents the oncoming straight-moving vehicles.

## Indicators for compliance threshold determination on highway
During the threshold selection process, in addition to excluding abnormal data, we calculated certain indicators to obtain the final

thresholds. The removal of abnormal data is described in the Supplementary Methods. The selection of the high-speed thresholds $t_{cl\_max}$ and $TTC_{cl\_min}$ is relatively straightforward. However, for determining the $d_{cl\_min}$ threshold, an optimization algorithm is used to determine the optimal threshold line.

The longitudinal and lateral RSS distances are calculated using parameter values from previous studies[47,48]. The longitudinal and lateral RSS distances can be calculated as follows:

$$d_{lonRSS} = \left[ 0.458v_r + 0.251 + \frac{(v_r + 0.978)^2}{4.272} - \frac{v_f^2}{15.250} \right]_+ \quad (19)$$

$$d_{latRSS} = 0.2 + \left[ \frac{2v_1 + 0.337}{2} \times 0.496 + \frac{(v_1 + 0.337)^2}{0.900} \right. \\ \left. - \left( \frac{2v_2 - 0.337}{2} \times 0.496 - \frac{(v_2 - 0.337)^2}{0.900} \right) \right]_+ \quad (20)$$

where $v_r$ and $v_f$ represent the velocities of the rear and forward vehicles, respectively, and $v_1$ and $v_2$ represent the lateral velocities of the rear and forward vehicles, respectively.

The fitness function of the threshold line optimization is defined as follows:

$$J = \max_f (\text{cost}) \quad s.t. \quad P_{FP}(f) \le r_{FP} \quad (21)$$

$$\text{cost} = N_{TP} \cdot Q + N_{TN} \quad (22)$$

where $f$ represents the threshold fitting curve of $d_{cl\_min}$ and $f = a \cdot x + b$, with $a$, and $b$ being undetermined coefficients. $P_{FP}$ represents the false-positive rate, and $r_{FP}$ represents the maximum allowable false-positive rate. The cost function is defined as the weighted sum of $N_{TP}$ and $N_{TN}$. $N_{TP}$ represents the number of true positives, $N_{TN}$ represents the number of true negatives, $Q$ is the weighting coefficients.

### Indicators for compliance threshold determination at intersection

In the scenario of a left-turn conflict with a straight-moving vehicle, it is difficult for the turning vehicle to generate as much displacement as possible in the forward direction of the straight-moving vehicle within the virtual lane, as defined in RSS. To address this issue, we adjusted RSS to eliminate the travel displacement of the leading vehicle. The adaptive RSS distance $d_{aRSS}$ is defined as follow:

$$d_{aRSS} = v_s\rho + 0.5a_{max,accel}\rho^2 - \frac{(v_s + \rho a_{max,accel})^2}{2a_{min,brake}} \quad (23)$$

where $v_s$ represents the velocity of the straight-moving vehicle when the trigger is first activated, $\rho$ represents the reaction time, which is equal to 0.458 s[47]; and $a_{min,brake}$ represents the minimum braking deceleration, which is set to −0.7 m/s$^2$, i.e., the lower limit of deceleration for a straight-moving vehicle without interference.

The maximum acceleration $a_{max,accel}$ of a straight-moving vehicle when it crosses the stop line until a trigger is activated is defined as follows:

$$a_{max,accel} = \max(a(t)) \quad s.t. \quad t_s \le t \le t_{act} \quad (24)$$

where $t_s$ represents the time at which the straight-moving vehicle starts to overlap the stop line and $t_{act}$ represents the time at which the trigger is first activated.

The minimum acceleration $a_{min}$ of a straight-moving vehicle when the trigger is activated until the left-turning or straight-moving vehicle reaches the intersection point is defined as follows:

$$a_{min} = \min(a(t)) \quad s.t. \quad t_s \le t \le t_{pass} \quad (25)$$

where $t_{pass}$ represents the time at which the left-turning or straight-moving vehicles reach their trajectory intersection point.

The intersection points $(x_{int}, y_{int})$ can be calculated easily using the intersection point formula of the line segments. Using the coordinates of this intersection point and the states of the left-turning and straight-moving vehicles in the dataset, $TTI_{left}$ and $TTI_{str}$ when the trigger is active can be calculated as follows:

$$TTI = \frac{l_{traj}}{\sqrt{(vx_{act}^2 + vy_{act}^2)}} \quad (26)$$

$$l_{traj} = \left( \sum_{k=1}^{N-2} \sqrt{(x_{k+1} - x_k)^2 + (y_{k+1} - y_k)^2} \right) \\ + \sqrt{(x_{int} - x_{N-1})^2 + (y_{int} - y_{N-1})^2} - \frac{l_{veh}}{2} \quad (27)$$

where $l_{traj}$ represents the length of the trajectory between the vehicle and the intersection point; $vx_{act}$ and $vy_{act}$ represent the longitudinal and lateral speeds, respectively, of the vehicle when the trigger starts to activate, $N$ represents the minimum number of sampling points for trajectories that contain intersection point; $l_{veh}$ represents the vehicle length.

## Data availability
The statistical results from the datasets generated in this study are available via Figshare under accession code https://doi.org/10.6084/m9.figshare.24372535[49]. It includes AD4CHE dataset, SIND dataset, Recorded scenarios, and Original traffic law. We provided the explanatory document about the dataset, along with some dataset segments of SIND dataset. The raw SIND dataset used to validate the method and analyze the thresholds is publicly available at https://github.com/SOTIF-AVLab/SinD. The raw AD4CHE dataset used to validate the method and perform threshold analysis is available under restricted access for the privacy requirement of the owner, access can be obtained by applying at https://auto.dji.com/cn/ad4che-dataset. Recorded scenarios include the input and output data of the Field test. Original traffic law includes the original traffic laws and the subdivided version. Source Data is available as a Source Data file and has also been deposited in Figshare[49]. Source data are provided with this paper.

## Code availability
Supplementary Code 1 includes codes for the AD4CHE analysis, SIND analysis, Online monitor program. AD4CHE analysis and SIND analysis contain code and output data for the dataset analysis. Online monitor program is a MATLAB version of the monitoring of articles related highway. These are also available in Figshare[49].

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

## Acknowledgements

This work was supported by the National key R&D Program of China: 2022YFB2503003 (Hong Wang), and the National Science Foundation of China Project: 52072215 (Hong Wang) and U1964203 (Jun Li).

## Author contributions

H.W. and W.Y. led the research program and proposed the initial concept of online legal monitor for autonomous driving. W.Y., C.Z. and H.W. developed the algorithms, performed the simulation tests, prepared the results and wrote the paper. D.Z. provided the technical approach in traffic law formalization and polished the paper. H.W., D.Z. and J.L. provided solid suggestions for the threshold selection method. H.W., J.L. and X.H. helped drone data processing. X.M., W.W. and Y.Y. provide the field test support. W.Y. and C.Z. contributed equally to this work. H.W. takes responsibility for the organization of the overall paper and approved the submission.

## Competing interests

The authors declare no competing interests.
