## [Peer Review File · Nature Communications]

Online Legal Driving Behavior Monitoring for Self-driving VehiclesREVIEWER COMMENTS

Reviewer #1 (Remarks to the Author):

The paper is a timely, interesting and well written exploration of how automated vehicles should interpret and respond to traffic laws since they have been written for human drivers. It discusses the potential role of an online monitor of rule compliance to support better decision making by automated vehicles.

The paper would benefit from a careful review of spellings and typos.

The conclusion could also discuss how to involve a variety of stakeholders (including the public) in determining behavioural characteristics for automated vehicles (including rule compliance, setting thresholds etc.) – and that the online monitor could be a very helpful way to engage in this discussion in an objective manner.

Overall, I think the paper is worthy of publication subject to some minor revisions as described (above and below)

Specific comments:

Abstract (and throughout) – The preference in the road safety sector is now not to use the term 'accident' but to refer to an 'incident', 'crash' or 'collision'

P1. 1. Introduction, para 1 – I don't think traffic laws are 'essential' for safety but they help to encourage safe driving and help to make behaviours more predictable for all road users.

P1. 1. Introduction, para 1 – "This requires AVs to follow the traffic laws strictly in the same way as human drivers follow them" – I'm pretty sure people don't 'strictly' follow traffic laws. Also I'm not sure AVs will have to follow traffic laws in exactly the same way that people do, provided that they behave in ways that are safe and predictable.

P1. 1. Introduction, para 2 – "...it is difficult for AVs to understand natural language..." – I think it is anthropomorphism to suggest that an AV might 'understands' natural language. Might be better to frame this as "...it is difficult for AVs to comply with traffic rules written in natural language in all circumstances...". It may also be worth noting that, in the event of an incident, the extent to which a human has correctly understood and applied traffic law is usually decided by a judge and jury, so there is case law to help determine what might be considered the correct behaviours.

P1. 1. Introduction, para 2 – "Specifically, AVs can only interact with digital information that has precise meanings." Is this still true in the age of AI, LLMs etc...?

P3. 2. Traffic laws and Constraints, para 1 – "...all geared towards ensuring the safety of both drivers and pedestrians." I think this would be better phrased as: "...all intended to support the safety of road users." (this then covers cyclists, horse riders etc. – all of which are included in traffic rules).

P3. 2. Traffic laws and Constraints, para 1 – "reflect the unite" – this should be "reflect the unity".

P4. 2. Traffic laws and Constraints, para 3 – "actions are clearly define while" – this should be "actions are clearly defined while".

P5. 3. Traffic Law digitization, fig. 3 – "Law breaked" – should be "Law broken"

Reviewer #2 (Remarks to the Author):

1. Interesting research! This study proposed an online traffic law violation monitor method. However, the theoretical and practical contribution of this study is limited because of the unsound methodology and the unreliable data process. The authors should state it clear.

2. P2, L3 in the title of Fig.1, it should be "two figures".

3. The part of "Compliance thresholds" should be the core part in this study, but the paper lacked the detail process.

4. The authors need to state clearly what is "online monitoring" and what is "offline monitoring".

Reviewer #3 (Remarks to the Author):

I thank the authors for submitting this work. It is an interesting paper in an emerging and important area (application of traffic laws to AVs). As far as I can tell, the results are valid and this manuscript takes the interesting step of making this abstract area logic more relevant by applying it to a real world problem (translating specific Chinese laws and applying it to parts of real-world traffic data sets).

The translation of traffic laws of machines such as automated vehicles or automated enforcement mechanisms is an unsolved problem and one of significant social and industry interest. As far as the novelty and advancement that this paper represents, let us look at the three claims the paper makes for its novelty: 1) architecture for online validation, 2) formalization of specific traffic laws, and 3) application to real world data sets

I did not feel like the first two claims represented significant advances. There is already a robust literature on traffic law formalization, much of which is not reviewed by the authors. I would recommend that the authors review the work of the Althoff group, as well as papers written by Calin Belta and Noushin Mehdipour on this topic (which includes papers from the Rulebooks group at nuTonomy/Motional). Therefore, I do not see the overall discussion of issues in formalizing traffic laws and the methodologies for doing so as contributing something significantly new for the methods that would be used in an online monitor of vehicle behavior (authors should read Bin-Nun 2022).

The second claim is that this paper formalizes several Chinese traffic laws using metric temporal logic (MTL). As far as I am aware, no one has formalized these specific traffic laws at all, or using MTL. However, researchers have previously formalized other traffic laws using MTL and other temporal logics. I am not convinced that, on its own, applying known methodologies to additional traffic laws rises to the level of novelty of interest to the general audience of this journal.

The most interesting part of this paper was the application of monitoring to a naturalistic data sets (SinD and AD4CHE) obtained by drones. The automated detection of a broad range of traffic law violations on a dataset has not been done yet. Therefore, I read the results with great interest and I think this is the best claim of the paper to a novel contribution and I would recommend expanding this analysis and centering the paper on this. In particular, I would like the authors to discuss these results and what we learn from them (is the number and distribution of violations surprising? Are they very sensitive to parameter and formalization choice? Are the results reasonable?)

A few other significant comments:

The authors should significantly expand their literature review and better situate their contribution.

The authors should better explain why those chose the 5 specific traffic laws they did.

The authors should better introduce the data sets and their limitations

Some of the traffic laws require significant research and analysis to set the compliance threshold and the authors make interesting and valiant attempts to do so. This is novel work, as there is little existing precedent. I think the authors made many interesting observations in this section, but they could set some more general principles and look for existing precedents (for example, the international regulations UNECE R157 and R79 set out parameters for following too closely, when lane changes interfere with cars in target lane, and how long lane changes should last, all of which this authors touch on without reference.

Overall comments:

I think the application of traffic laws to AVs is an important area and this manuscript makes contributions, particularly in the exercise of trying to practically implement a traffic law violation detector and running it on an actual dataset. The results, obtained by running the detectors on real-world traffic data, have the potential to be quite interesting.

Right now, the manuscript is much more focused on the underlying MTL methodology, which is less novel. To the extent this remains the focus of the paper, it may represent an incremental advance which belongs in a more specialized journal.

To publish in this journal, I would urge the authors to expand, center, and put more focus on the results obtained from real world data and their broader implications. This will also necessitate grappling more with the formalization choices and the sensitivity of results to those choices. Such an expansion, particularly if it is brought into contact with the broader literature on driving behavior and safety, has potential to be a significant contribution.

RESPONSES TO REVIEWERS' COMMENTS

Manuscript ID: NCOMMS-23-20905

Manuscript Title: Online Legal Driving Behavior Monitoring for Self-driving Vehicles

Authors: Wenhao Yu[†], Chengxiang Zhao[†], Jiabin Liu , Yingkai Yang, Xiaohan Ma, Jun Li, Weida Wang, Hong Wang[✉], Xiaosong Hu[✉], Ding Zhao

REVIEWER COMMENTS

Reviewer #1 (Remarks to the Author):

The paper is a timely, interesting and well written exploration of how automated vehicles should interpret and respond to traffic laws since they have been written for human drivers. It discusses the potential role of an online monitor of rule compliance to support better decision making by automated vehicles.

The paper would benefit from a careful review of spellings and typos.

The conclusion could also discuss how to involve a variety of stakeholders (including the public) in determining behavioural characteristics for automated vehicles (including rule compliance, setting thresholds etc.) – and that the online monitor could be a very helpful way to engage in this discussion in an objective manner.

Overall, I think the paper is worthy of publication subject to some minor revisions as described (above and below)

Specific comments:

Abstract (and throughout) – The preference in the road safety sector is now not to use the term ‘accident’ but to refer to an ‘incident’, ‘crash’ or ‘collision’

P1. 1. Introduction, para 1 – I don’t think traffic laws are ‘essential’ for safety but they help to encourage safe driving and help to make behaviours more predictable for all road users.

P1. 1. Introduction, para 1 – “This requires AVs to follow the traffic laws strictly in the same way as human drivers follow them” – I’m pretty sure people don’t ‘strictly’ follow traffic laws. Also I’m not sure AVs will have to follow traffic laws in exactly the same way that people do, provided that they behave in ways that are safe and predictable.

P1. 1. Introduction, para 2 – “...it is difficult for AVs to understand natural language...” – I think it is anthropomorphism to suggest that an AV might ‘understands’ natural language. Might be better to frame this as “...it is difficult for AVs to comply with traffic rules written in natural language in all circumstances...”. It may also be worth noting that, in the event of an incident, the extent to which a human has correctly understood and applied traffic law is usually decided by a judge and jury, so there is case law to help determine what might be considered the correct behaviours.

P1. 1. Introduction, para 2 – “Specifically, AVs can only interact with digital information that has precise meanings.” Is this still true in the age of AI, LLMs etc...?

P3. 2. Traffic laws and Constraints, para 1 – “...all geared towards ensuring the safety of both drivers and pedestrians.” I think this would be better phrased as: “...all intended to support the safety of road users.” (this then covers cyclists, horse riders etc. – all of which are included in

traffic rules).

P3. 2. Traffic laws and Constraints, para 1 – “reflect the unite” – this should be “reflect the unity”.

P4. 2. Traffic laws and Constraints, para 3 – “actions are clearly define while” – this should be “actions are clearly defined while”.

P5. 3. Traffic Law digitization, fig. 3 – “Law breaked” – should be “Law broken”

Reviewer #2 (Remarks to the Author):

1. Interesting research! This study proposed an online traffic law violation monitor method.

However, the theoretical and practical contribution of this study is limited because of the unsound methodology and the unreliable data process. The authors should state it clear.

2. P2, L3 in the title of Fig.1, it should be "two figures".

3. The part of "Compliance thresholds" should be the core part in this study, but the paper lacked the detail process.

4. The authors need to state clearly what is "online monitoring" and what is "offline monitoring".

Reviewer #3 (Remarks to the Author):

I thank the authors for submitting this work. It is an interesting paper in an emerging and important area (application of traffic las to AVs). As far as I can tell, the results are valid and this manuscript takes the interesting step of making this abstract area logic more relevant by applying it to a real world problem (translating specific Chinese laws and applying it to parts of real-world traffic data sets).

The translation of traffic laws of machines such as automated vehicles or automated enforcement mechanisms is an unsolved problem and one of significant social and industry interest. As far as the novelty and advancement that this paper represents, let us look at the three claims the paper makes for its novelty: 1) architecture for online validation, 2) formalization of specific traffic laws, and 3) application to real world data sets

I did not feel like the first two claims represented significant advances. There is already a robust literature on traffic law formalization, much of which is not reviewed by the authors. I would recommend that the authors review the work of the Althoff group, as well as papers written by Calin Belta and Noushin Mehdipour on this topic (which includes papers from the Rulebooks group at nuTonomy/Motional). Therefore, I do not see the overall discussion of issues in formalizing

traffic laws and the methodologies for doing so as contributing something significantly new for the methods that would be used in an online monitor of vehicle behavior (authors should read Bin-Nun 2022).

The second claim is that this paper formalizes several Chinese traffic laws using metric temporal logic (MTL). As far as I am aware, no one has formalized these specific traffic laws at all, or using MTL. However, researchers have previously formalized other traffic laws using MTL and other temporal logics. I am not convinced that, on its own, applying known methodologies to additional traffic laws rises to the level of novelty of interest to the general audience of this journal.

The most interesting part of this paper was the application of monitoring to a naturalistic data sets (SinD and AD4CHE) obtained by drones. The automated detection of a broad range of traffic law violations on a dataset has not been done yet. Therefore, I read the results with great interest and I think this is the best claim of the paper to a novel contribution and I would recommend expanding this analysis and centering the paper on this. In particular, I would like the authors to discuss these results and what we learn from them (is the number and distribution of violations surprising? Are they very sensitive to parameter and formalization choice? Are the results reasonable?)

A few other significant comments:

The authors should significantly expand their literature review and better situate their contribution.

The authors should better explain why those chose the 5 specific traffic laws they did.

The authors should better introduce the data sets and their limitations

Some of the traffic laws require significant research and analysis to set the compliance threshold and the authors make interesting and valiant attempts to do so. This is novel work, as there is little existing precedent. I think the authors made many interesting observations in this section, but they could set some more general principles and look for existing precedents (for example, the international regulations UNECE R157 and R79 set out parameters for following to closely, when lane changes interfere with cars in target lane, and how long lane changes should last, all of which this authors touch on without reference.

Overall comments:

I think the application of traffic laws to AVs is an important area and this manuscript makes contributions, particularly in the exercise of trying to practically implement a traffic law violation detector and running it on an actual dataset. The results, obtained by running the detectors on real-world traffic data, have the potential to be quite interesting.

Right now, the manuscript is much more focused on the underlying MTL methodology, which is less novel. To the extent this remains the focus of the paper, it may represent an incremental advance which belongs in a more specialized journal.

To publish in this journal, I would urge the authors to expand, center, and put more focus on the results obtained from real world data and their broader implications. This will also necessitate grappling more with the formalization choices and the sensitivity of results to those choices. Such an expansion, particularly if it is brought into contact with the broader literature on driving behavior and safety, has potential to be a significant contribution.

Response to the reviewers' comments

The authors would like to express our sincere appreciation for their scrupulous review and constructive comments. In response to the feedback from all the reviewers, we have undertaken substantial revisions, both in terms of structure and content, resulting in the creation of the current version of the manuscript. After further modification, the whole manuscript has been polished by native speakers.

In the following, we will provide a point-to-point response to all comments. Normally they are in blue after all the reviewers' comments and suggestions, and citations from the manuscript are also marked in blue. Thank you for the reviewers' comments again. It has encouraged us to improve our manuscript further.

Reviewer 1

The paper is a timely, interesting and well written exploration of how automated vehicles should interpret and respond to traffic laws since they have been written for human drivers. It discusses the potential role of an online monitor of rule compliance to support better decision making by automated vehicles. The paper would benefit from a careful review of spellings and typos. The conclusion could also discuss how to involve a variety of stakeholders (including the public) in determining behavioural characteristics for automated vehicles (including rule compliance, setting thresholds etc.) - and that the online monitor could be a very helpful way to engage in this discussion in an objective manner. Overall, I think the paper is worthy of publication subject to some minor revisions as described (above and below).

Response:

We thank the reviewer for the scrupulous review and positive feedback. After careful revision, the revised manuscript and supplementary information have been polished by native speakers to avoid spelling mistakes and typos. Regarding the online monitor, we have reframed the paper and rewritten both the Introduction and Discussion sections as follows: “Legal driving is a prerequisite for the widespread adoption of self-driving vehicles (SVs) to ensure the safety of future transportation [1]. Independent online monitoring of the driving behavior of SVs is not only an essential means for government regulation of autonomous driving, such as provides substantial evidence for the traceability of traffic incidents, but also can provide warnings of violations to autonomous driving algorithms, helping improve their compliance with regulations.”, “In this work, a trigger-based hierarchical online monitor was proposed, along with driving behavior-based ambiguous compliance thresholds determination, to ensure the rational and fact-based legal judgments align with common understanding. This work has three notable advantages. First, a trigger-based hierarchical structure

for the online monitor only triggers the articles that should be monitored and utilizes the driving behavior to increase the rationality of monitoring. Second, ambiguous human-oriented traffic laws can be implemented into this online monitor via compliance threshold determination utilizing general principles. Finally, the monitor results based on the drone datasets and field test demonstrate the feasibility and rationality of the proposed online legal driving behavior monitor. However, the process of establishing precise, specific compliance thresholds for evaluating complex behaviors of traffic participants requires further data validation. Constructing and updating joint distribution parameters [43, 44] for critical behavioral variables and deriving specific probabilities of behavioral violations from a larger dataset of real-world violations may yield more robust results. Equally important is the need for feedback mechanisms involving government agencies and various stakeholders to collaboratively define the ultimate compliance threshold for widespread implementation in self-driving vehicles. It is essential to acknowledge that traditional traffic laws, designed for human drivers, may not fully satisfy the requirements of self-driving vehicles in the future. Leveraging the quantitative data processing capabilities of self-driving vehicles, we have the opportunity to develop more detailed traffic regulations tailored specifically to them. These regulations can strike a balance between safety, efficiency, and compliance in complex scenarios while taking into account human factors and ethical considerations, thereby promoting the legal implementation of self-driving vehicles.”

Comment 1:

Abstract (and throughout) – The preference in the road safety sector is now not to use the term ‘accident’ but to refer to an ‘incident’, ‘crash’ or ‘collision’.

Response:

We agree with the reviewer and accept the comments. The word ‘accident’ has been changed into ‘incident’ in the whole manuscript. Specifically, we have changed the sentence to “**Independent** online monitoring of the driving behavior of SVs is not only an essential means for government regulation of autonomous driving, such as provides substantial evidence for the traceability of traffic incidents, but also can provide warnings of violations to autonomous driving algorithms, helping improve their compliance with regulations.”

Comment 2:

Pl. 1. Introduction, para 1 – I don’t think traffic laws are ‘essential’ for safety but they help to encourage safe driving and help to make behaviours more predictable for all road users.

Response:

Thanks for the reviewers’ suggestion. We have rewritten the Introduction section of the paper and removed the words that may make the paper sound grandiose. The revision is as follows: “**Legal**

driving is a prerequisite for the widespread adoption of self-driving vehicles (SVs) to ensure the safety of future transportation.”

Comment 3:

Pl. 1. Introduction, para 1 – “This requires AVs to follow the traffic laws strictly in the same way as human drivers follow them” – I’m pretty sure people don’t ‘strictly’ follow traffic laws. Also I’m not sure AVs will have to follow traffic laws in exactly the same way that people do, provided that they behave in ways that are safe and predictable.

Response:

Again, we agree with the reviewer. It is certain that people don’t “strictly” follow the traffic laws. In real-life scenarios, some of the traffic law articles are treated as soft constraints, as breaking those articles will get no punishment until an incident occurs. But the government and other traffic participants are surely not expecting that behavior as it sacrifices others’ safety. We can use an online monitor (government monitoring level) and an online monitoring result-based traffic law compliance decision-making algorithm (technical level) in AVs to prevent this event from happening and achieve a safer traffic environment. The revision is shown in Abstract and Introduction section as follows: “Defined traffic laws must be respected by all vehicles when driving on the road, including self-driving vehicles without human drivers.”, “Legal driving is a prerequisite for the widespread adoption of self-driving vehicles (SVs) to ensure the safety of future transportation [1]. Independent online monitoring of the driving behavior of SVs is not only an essential means for government regulation of autonomous driving, such as provides substantial evidence for the traceability of traffic incidents, but also can provide warnings of violations to autonomous driving algorithms, helping improve their compliance with regulations.”

Comment 4:

Pl. 1. Introduction, para 2 – “...it is difficult for AVs to understand natural language...” – I think it is anthropomorphism to suggest that an AV might ‘understands’ natural language. Might be better to frame this as “...it is difficult for AVs to comply with traffic rules written in natural language in all circumstances...”. It may also be worth noting that, in the event of an incident, the extent to which a human has correctly understood and applied traffic law is usually decided by a judge and jury, so there is case law to help determine what might be considered the correct behaviours.

Response:

We fully agree with that it is hard to assert whether AVs can “understand” traffic laws. The key problem is as you mentioned that it is difficult for AVs to comply with traffic rules written in natural language in all circumstances. We totally agree that case law is helpful to determine what might be the considered correct behaviours. The revised revision is as follows to avoid ambiguous statements: “Currently, human-oriented traffic laws contain numerous ambiguous expressions, leading to varying

interpretations by enterprises, and consequently, divergent behaviors in autonomous-vehicle (AV) systems [2-4]. The implementation of human-oriented traffic laws for real-time vehicle driving remains a challenge.”

Comment 5:

P1. 1. Introduction, para 2 – “Specifically, AVs can only interact with digital information that has precise meanings.” Is this still true in the age of AI, LLMs etc...?

Response:

Again, we agree with the review that it’s not accurate to say AVs can only interact with digital information that has precise meanings especially when AI and LLM have come to reality and demonstrate the great ability of language recognition and reasoning in some research fields. We have removed this statement.

Comment 6:

P3. 2. Traffic laws and Constraints, para 1 – “...all geared towards ensuring the safety of both drivers and pedestrians.” I think this would be better phrased as: “...all intended to support the safety of road users.” (this then covers cyclists, horse riders etc. – all of which are included in traffic rules).

Response:

We fully agree with the reviewer. In the revised version we have removed the statement and used “traffic participants” to phrase the road user in the whole paper. Relevant modifications are as follows “Additionally, Bogdoll *et al.* [35] attempted to determine the threshold for the following distance based on the Waymo dataset, which accommodates variations in the behaviors of different traffic participants.” , “When applied in the open environment, other methods may encounter challenges in triggering correct law article monitoring in specific scenarios or in aligning with the behavior of the majority of traffic participants.” , “This allows the continuous differentiation of the operational environment and behaviors of the surrounding traffic participants encountered by the ego vehicle, facilitating the correct monitoring of relevant law articles.” , “By analyzing the behavior of most traffic participants, compliance thresholds aligned with real-world behavior are obtained.” , “Owing to the numerous factors that influence the behavior of each traffic participant, even when mathematical combinations of key factors are employed, eliminating the impact of other factors remains difficult.” , “The SIND dataset lasted for approximately 957 mins, encompassing 30,953 trajectories, including 53 records and involving 7 types of traffic participants (cars, buses, trucks, motorcycles, bicycles, tricycles, and pedestrians), as shown in Figs.3(e-h)” , “All the necessary information was transmitted to the online monitor by the designed data bus in a dataset sampling time

sequence that contained ego vehicle parameters, traffic signs, traffic participants, and map data.” , “However, the process of establishing precise, specific compliance thresholds for evaluating complex behaviors of traffic participants requires further data validation.” , “The information of the other traffic participants is represented in the vehicle coordinate system.”

Comment 7:

P3. 2. Traffic laws and Constraints, para 1 – “reflect the unite” – this should be “reflect the unity”.

Response:

Thanks for the reviewer’s comments. In the revised version this statement is removed, and the relevant revision is as follows: “The behaviors constrained by the traffic laws and the meanings of the constraints are generally consistent. This consistency makes it possible to utilize a formalized framework of laws to solve the formalization problem of different laws in different regions.”

Comment 8:

P4. 2. Traffic laws and Constraints, para 3 – “actions are clearly define while” – this should be “actions are clearly defined while”.

Response:

Thanks for the reviewer’s meticulous comment. We have revised the framework of the paper to focus on threshold analysis and online monitoring results, thus we have removed the original detailed discussion on the types of traffic laws and constraints in the revised main manuscript.

Comment 9:

P5. 3. Traffic Law digitization, fig. 3 – “Law breaked” – should be “Law broken”.

Response:

Thank you for the reviewer’s meticulous suggestion. We have replaced “Law broken” with “Violation” for clearer expression as shown in the bottom part in Fig.1 Online traffic-law violation monitor for AVs in the revised manuscript.

Fig. 1 Online traffic-law violation monitor for AVs

Fig. 1 Online traffic-law violation monitor for AVs. This monitoring system is capable of deployment within the SV and monitors the SV's adherence to traffic laws. It receives real-time data from the AV system and provides continuous monitoring results of the ego vehicle. The monitoring system has a trigger-based hierarchical architecture that ensures structural integrity (e.g., drive on lane line (a2) \subseteq make lane-change (b1 & b3) \subseteq overtake (c1) or keep lateral distance(a3) \subseteq encounter (b2)), which enhances the rationality of the monitoring results and simplifies maintenance in later stages.

Reviewer 2

Comment 1:

Interesting research! This study proposed an online traffic law violation monitor method. However, the theoretical and practical contribution of this study is limited because of the unsound methodology and the unreliable data process. The authors should state it clear.

Response:

We sincerely thank for the reviewer's time and patient review. According to the comments, we have reframed our main contributions to place a greater emphasis on an expanded and centralized compliance threshold analysis: (1) Trigger-based hierarchical online monitor architecture, (2) Fact-based logical judgment and data-based thresholds, (3) Sensitivity analysis of compliance thresholds.

Additionally, two general principles were proposed to determine the ambiguous compliance thresholds. 1) No Crashes: For safety, there should be no crashes with other vehicles. This is ensured through safety-related indices such as the Time to Collision, Risk Sensitive Safety and other kinematics models, 2) No Changes: The ego vehicle's behavior should not be the reason to cause the change of other vehicles' behavior, which can be established by assessing nearby vehicles whose trajectories intersect with the ego vehicle. If no significant braking or steering responses are observed, we assume that the surrounding vehicle is not affected by the ego vehicle.

Our analysis is grounded in extensive data validation and has been seamlessly integrated into our monitoring system using authentic driving behavior datasets and rigorous vehicle field tests. To augment our dataset, we expanded upon the original AD4CHE and SIND dataset by collecting additional intersection data from three distinct cities across diverse regions (Changchun, Xi'an, Chongqing), spanning a cumulative distance of 2300 kilometers. This enriched dataset forms the foundation for our compliance threshold analysis. Furthermore, we conducted a real vehicle test to showcase the real-time capabilities of our monitoring system. These efforts have substantially fortified the robustness of our results and the persuasiveness of our conclusions. For comprehensive information on these enhancements, please refer to the Introduction, Thresholds Analysis, Monitoring Results, and Field Test sections in the revised manuscript.

Comment 2:

P2, L3 in the title of Fig.1, it should be "two figures".

Response:

We agree with the reviewer and the revised figure is shown in Fig.1 as follows,

Fig. 1 Online traffic-law violation monitor for AVs. This monitoring system is capable of deployment within the SV and monitors the SV's adherence to traffic laws. It receives real-time data from the AV system and provides continuous monitoring results of the ego vehicle. The monitoring system has a trigger-based hierarchical architecture that ensures structural integrity (e.g., drive on lane line (a2) \subseteq make lane-change (b1 & b3) \subseteq overtake (c1) or keep lateral distance(a3) \subseteq encounter (b2)), which enhances the rationality of the monitoring results and simplifies maintenance in later stages.

Comment 3:

The part of "Compliance thresholds" should be the core part in this study, but the paper lacked the detail process.

Response:

We fully agree with the reviewer that the part of "Compliance thresholds" should be the core part in this study. We have emphasized the compliance threshold in the revised version, including the selection principles, the involved dataset, the trajectory filtering methods, and the calculations are all stated in detail in the Thresholds Analysis section. We proposed two general principles to determine the ambiguous compliance thresholds. 1) *No Crashes*: For safety, there should be no crashes with other vehicles. This is ensured through safety-related indices such as the Time to Collision, Risk Sensitive Safety and other kinematics models. 2) *No Changes*: The ego vehicle's behavior should not be the reason to cause the change of other vehicles' behavior, which can be established by assessing nearby vehicles whose trajectories intersect with the ego vehicle. If no significant braking or steering responses are observed, we assume that the surrounding vehicle is not affected by the ego vehicle.

Specifically, according to the selected articles, there are four thresholds in ambiguous expressions that need to be determined: 1) the maximum allowable time to drive on the lane line (t_{cl_max}) was determined to specify the expression “drive over or on the dividing line” in Article 82.6; 2) when making lane-change, the minimum allowable TTC with the preceding vehicle (TTC_{cl_min}) and 3) the minimum allowable distance from the rear vehicle in the target lane (d_{cl_min}) were determined to specify the expression “not impede” in Article 44; 4) the minimum allowable time difference between a left-turn vehicle and a straight-moving vehicle to the intersection point (TTI_{diff_min}) was determined to specify the expression “not interfere” in Article 38.2. Through the analyzation, the specific selection of compliance threshold is shown as follow, more detail can be found in Threshold Analysis and Sensitivity analysis sections in the revised manuscript.

- 1) t_{cl_max} : The statistical results followed a reverse Gaussian distribution with fitted parameters of $\mu=2.791$ and $\lambda=20.689$. Vehicles crossed lane lines for durations up to 6 s in 99.04% of the cases. Therefore, t_{cl_max} was determined to be 6 s, ensuring that standard lane-change maneuvers occurred within this specified time, as shown in Fig.4c in the revised manuscript.

Fig. 4c Distribution of the time to drive on the lane line

- 2) TTC_{cl_min} :The fitting curve represents the relationship between TTC_{ini} and t_{cl} that most drivers follow during the lane-change period. The TTC_{ini} at the intersection point of the fitted curve and the limiting value is 1.8 s, indicating that most drivers maintained a TTC_{cl_min} of 1.8 s with the preceding vehicle during the lane-change maneuver, as shown in Fig.4d in the revised manuscript.

Fig. 4d The ratio of t_{cl} to TTC_{ini} in different TTC_{ini} data points and fitting curve

3) d_{cl_min} : the d_{cl_min} value was determined as follows, as shown in Fig.5h in the revised manuscript.

$$d_{cl_min} = \begin{cases} 50, & \Delta v < -10.7 \\ -3.4\Delta v + 13.6, & -10.7 \leq \Delta v \leq 4 \\ 0, & \Delta v > 4 \end{cases}$$

where Δv represents the speed difference in m/s between the ego vehicle and RVTL at each sample time when the trigger activated.

Fig. 5h Optimal threshold line and RVTL' $d_{cl} - \Delta \bar{v}$ data points marked with different compliance states when the trigger is first activated in selected instances

4) TTI_{diff_min} : Through sensitivity analysis, the final selection of TTI_{diff_min} was 3.4 s. It is worth noting that, the slope of threshold line of d_{cl_min} indicates a time relationship between the ego vehicle and RVTL is also 3.4s, as shown in Fig.6e in the revised manuscript.

Fig. 6e States of straight-moving vehicles when left-turning vehicles pass the conflict zone first

Comment 4:

The authors need to state clearly what is "online monitoring" and what is "offline monitoring".

Response:

We agree with the reviewer the online monitoring and offline monitoring should be clearly stated. Online monitoring and offline monitoring have different scopes of application, different input information, and results variability. Detailed information can be found in Supplementary Information: “Offline monitoring is a law violation monitoring that judges the compliance of the driving behaviors of one or more vehicles in the whole scenario by obtaining the whole-period vehicles’ behavior information. In contrast, online monitoring is a law violation monitoring that judges the compliance of driving behaviors of the ego vehicle or all vehicles from the start of the monitoring process to the current time using the observed or collected data. It should be noted that different purposes of online monitoring show certain differences. Among online monitoring, three kinds of purposes are proposed 1) fact-based monitoring, 2) decision-based monitoring, and 3) prediction-based monitoring. The choice between online monitoring and offline monitoring will result in different formalization procedures and result formulas. Each kind of purpose is comprised in detail as Supplementary Table 2.

Table.2 The comparisons of different monitoring

Comparison	Offline monitoring	Online monitoring		
		Fact-based law violation monitoring	Decision-based law violation monitoring	Prediction-based law violation monitoring
Application	1). Used for offline evaluation of SV tasks of a vehicle; 2). Used in the roadside equipment to provide third-party monitoring for vehicle behaviors	1). Used at an SV for online monitor of SV's behaviors and division of accident responsibility; 2). Used in the roadside equipment for online third-party monitor of vehicle behavior violations	Used in a vehicle for online management of autonomous driving behaviors and online interference response for illegal decision-making	Used in a vehicle for online management of autonomous driving behaviors and online interference response for possible violations
Information	Ego vehicle: all time periods of all data Other vehicles: all time periods of all data	Ego vehicle: current and past periods of data Other vehicles: current and past periods of collected data	Ego vehicle: current and past periods of data and division-making data Other vehicles: current and past periods of observable data	Ego vehicle: current and past periods of data and division-making data Other vehicles: current and past periods of collected data and the corresponding future prediction data
Overall judgment difficulty	Easy	Hard	Medium	Medium
Results	Unchangeable	Unchangeable	Varying with decisions	Varying with decisions, control outputs and others’ behaviors
Decision friendly	None	Bad	Good	Best

In Supplementary Table 2, online monitoring is divided into three types according to the information used for monitoring. The fact-based monitoring uses historical and current vehicle behavior data. In this type of monitoring, once a law violation behavior is found, the violation result is an established fact that cannot be changed. Therefore, this type of monitoring can be used on both the vehicle side and the road equipment side for recording violation behaviors of the vehicles, and the result can be used as a reference for accident responsibility division. Besides historical and current vehicle behavior data, when the ego vehicle's decision data are involved, online monitoring is given the ability to "foresee" the future actions of the ego vehicle. This type of monitoring is regarded as decision-based law violation monitoring, and it can be used only for ego vehicles. However, this monitoring type can tell whether the ego vehicle will break the traffic law if following the current decision. Also, the decision-making system can read the monitor's output to adjust its decision to comply with the traffic law. Therefore, the monitoring result changes with the decision of the ego vehicle. Furthermore, the traffic law restrains the relationships between traffic participants. Thus, if it is required that the monitoring result has the best law compliance guiding significance, the prediction behaviors of other participants should also be considered, and this represents the prediction-based law violation monitoring. This monitoring type combines historical and current vehicle behavior data with the ego vehicle's decision and perception data to judge the law compliance of the decision in the prediction range. Owing to its heavy reliance on prediction, the monitoring result is unstable and varies with the decisions, other participants' behaviors and their predictions. However, this monitoring type is the most decision-friendly monitoring, and its result gives the best advance quantity to adjust the decision.

If the monitored vehicle is a white box for offline monitoring or only in certain given scenarios, it will be easy to select law monitoring algorithms to determine behavior violations for vehicle decisions or scenario types. However, when facing a black box vehicle and in a free-run situation, performing online monitoring is relatively challenging because it is necessary to estimate a vehicle's next action on the whole trajectory and which law is convenient for a particular case. Therefore, without the whole-trajectory data and using only past and current data that the ego vehicle collected, it is difficult to monitor behavior violations for the fact-based monitor because it is required to set more judgment conditions to determine which law is suitable for the current scenario. Furthermore, right-of-way monitoring is even more challenging because other traffic participants are involved. These participants' behaviors can lead to a situation where much more judgment conditions need to be discussed, and more thresholds should be considered. By using the whole-trajectory data of all participants or the ego vehicle's decision and prediction data, the monitor task will become easier to perform because the future data can reduce the condition classification discussions. This is the main reason the fact-based monitor is the most challenging to achieve."

[1] Bin-Nun, A. Y., Derler, P., Mehdipour, N. & Tebbens, R. D. How should autonomous vehicles drive? policy, methodological, and social considerations for designing a driver. *Humanities and social sciences communications* **9**, 1–13 (2022).

[17] Mehdipour, N., Althoff, M., Tebbens, R. D. & Belta, C. Formal methods to comply with rules of the road in autonomous driving: State of the art and grand challenges. *Automatica* **152**, 110692 (2023).

Reviewer 3

I thank the authors for submitting this work. It is an interesting paper in an emerging and important area (application of traffic laws to AVs). As far as I can tell, the results are valid and this manuscript takes the interesting step of making this abstract area logic more relevant by applying it to a real world problem (translating specific Chinese laws and applying it to parts of real-world traffic data sets).

Response:

Thanks for the reviewer's positive feedback and great encouragement.

Comment 1:

The translation of traffic laws of machines such as automated vehicles or automated enforcement mechanisms is an unsolved problem and one of significant social and industry interest. As far as the novelty and advancement that this paper represents, let us look at the three claims the paper makes for its novelty: 1) architecture for online validation, 2) formalization of specific traffic laws, and 3) application to real world data sets.

I did not feel like the first two claims represented significant advances. There is already a robust literature on traffic law formalization, much of which is not reviewed by the authors. I would recommend that the authors review the work of the Althoff group, as well as papers written by Calin Belta and Noushin Mehdipour on this topic (which includes papers from the Rulebooks group at nuTonomy/Motional). Therefore, I do not see the overall discussion of issues in formalizing traffic laws and the methodologies for doing so as contributing something significantly new for the methods that would be used in an online monitor of vehicle behavior (authors should read Bin-Nun 2022).

The authors should significantly expand their literature review and better situate their contribution.

Response:

We totally agree with the reviewer and thanks for the meticulous comments. We have expanded our literature review. In this revision, we first analyzed three commonly adopted formalization methods and three purposes of formalizing regulations. We then discussed research related to the ambiguous threshold issues in traffic regulation formalization in existing studies. Based on the updated literature review, the contribution has been reframed as three aspects: (1) Trigger-based hierarchical online monitor architecture. This allows the continuous differentiation of the operational environment and behaviors of the surrounding traffic participants encountered by the ego vehicle, facilitating the correct monitoring of relevant law articles; (2) Fact-based logical judgment and data-

based thresholds. By analyzing the behavior of most traffic participants, compliance thresholds aligned with real-world behavior are obtained; (3) Sensitivity analysis of thresholds. Through a sensitivity analysis, compliance monitoring thresholds are fine-tuned to strike a balance between false negatives (non-compliance but judged as compliance) and false positives (compliance but judged as non-compliance).

The updated literature review is presented as follows: “In the field of autonomous driving, most regulation-related research begins with the formalization of regulations. The teams of Althoff and Bin-Nun made major pioneering contributions to the formalization of traffic laws and subsequent applications. In early studies, simple logic formulas were used to formalize traffic laws, such as first-order logic [5], deontic-order logic [6, 7], and high-order logic [8, 9]. However, these methods cannot describe the sequential nature of the typical driving behavior of traffic laws [10]. Recently, temporal logic-based methods such as linear temporal logic (LTL) [11] have gained traction because of their expressiveness in traffic-law representation. Extensions such as signal temporal logic (STL) [12, 13] are additionally equipped with legality robustness degree, and metric temporal logic (MTL) [14–16] can further specify intervals for property fulfillment. The primary applications of formalized traffic laws include monitoring, control synthesis, and formal verification [17]. Monitoring refers to checking the current or recorded driving behaviors of SVs violate traffic laws [11, 18]. The control synthesis aims to solve a vehicle controller to plan an optimal trajectory within traffic laws restrictions [19–21]. Formal verification aims to theoretically prove or ensure the legality of all possible behaviors of a given SV system [22–24]. Most applications focus on enhancing the compliance of SVs within the current traffic-law framework, and only a few are dedicated to offering independent and reliable sources of compliance monitoring data for government regulation and enterprise analysis. The latter goal requires that the monitor encompasses all traffic-law articles, operates continuously to cover all road sections without interfering with the AV system’s decisions, and provides rational judgments based on the genuine driving behaviors of vehicles. The understanding of ambiguous traffic laws varies significantly among individuals, and the key challenge in achieving rational judgments is the selection of thresholds for ambiguous articles, such as “The vehicle behind shall overtake from the left side of the vehicle in front after making sure that there is sufficient safe space” and “the vehicles making a turn may not interfere the vehicles and pedestrians that are let go straight forward”. Many researchers have attempted to select thresholds for ambiguous articles, and some researchers [25, 26] sought relevant guidance from suggestive documents, such as driver guides [27, 28]. These suggestions are often derived from previous driving experience. The prevailing approach for threshold analysis is based on theoretical models that specify thresholds using pre-designed models with kinematic principles. For example, the driver reaction model with the maximum brake distance [11, 12, 29] or the set-base prediction model [30–32], can

be used to specify the safe distance. Thresholds from models are usually conservative owing to strict constraints that are applicable for decision-making to ensure legality. Recently, studies performed by Belta et al. [33, 34] demonstrated the potential of constructing data-based models from datasets. Additionally, Bogdoll et al. [35] attempted to determine the threshold for the following distance based on the Waymo dataset, which accommodates variations in the behaviors of different traffic participants. However, the driving guidance cannot be utilized to determine whether a violation has occurred. Moreover, safety models tend to be conservative, making it difficult to blame aggressive drivers for not adhering to the safety model unless an incident has occurred. Thus, it is imperative to establish rational thresholds based on driving behavior to ensure that they align with the distribution of the majority of human drivers' behavior.”

Comment 2:

The second claim is that this paper formalizes several Chinese traffic laws using metric temporal logic (MTL). As far as I am aware, no one has formalized these specific traffic laws at all, or using MTL. However, researchers have previously formalized other traffic laws using MTL and other temporal logics. I am not convinced that, on its own, applying known methodologies to additional traffic laws rises to the level of novelty of interest to the general audience of this journal.

Response:

Indeed, prior research has undertaken the formalization of various traffic laws, often utilizing MTL and other temporal logics. It is acknowledged that applying established methodologies to additional traffic laws may not be considered as groundbreaking. Thus, the formalization of traffic law using MIT is not considered as one of the main contributions in the revised manuscript. To streamline the reader's comprehension of the process, the formalization content has been relocated to the Supplementary Information. In this revised version, our focus has shifted towards emphasizing the selection of thresholds and presenting the monitoring results. To enhance the innovation of the paper, we have adopted data-based thresholds that align with the behavior distribution of the majority of drivers. We have introduced a method for traffic law compliance judgment solely utilizing driving behavioral data. The paper includes examples and analysis of key thresholds related to ambiguous expressions, particularly on highways and at intersections. We have also outlined the principles and processes governing threshold selection. Additionally, to strengthen our work, we have gathered more data for dataset validation and conducted real vehicle tests to verify the rationality and real-time performance of our approach.

Comment 3:

The most interesting part of this paper was the application of monitoring to a naturalistic data sets (SinD and AD4CHE) obtained by drones. The automated detection of a broad range of traffic law violations on a dataset has not been done yet. Therefore, I read the results with great interest and I think this is the best claim of the paper to a novel contribution and I would recommend expanding this analysis and centering the paper on this. In particular, I would like the authors to discuss these results and what we learn from them (is the number and distribution of violations surprising? Are they very sensitive to parameter and formalization choice? Are the results reasonable?)

Response:

We agree with the reviewer that the most interesting part of this paper should be the application of monitoring to the SIND and AD4CHE dataset. To illustrate more persuading and robust monitor result, based on the original dataset, we collected more intersection data from three different cities (Changchun, Xi'an, Chongqing). The SIND dataset now encompassed four two-phase signalized intersections situated in different Chinese cities spanning 2,300 km. A detailed introduction of each city is in Fig.3. The SIND dataset now added 30 new data fragments, approximately 537mins, encompassing 17,990 trajectories, emphasizing the threshold selection and results parts. At the same time, we used all data fragments of AD4CHE in this round revision. The results are as follows in Monitoring results on datasets section in the revised manuscript.

“For the AD4CHE dataset, in vehicles with corresponding trigger condition activated, there are a total of 18017 vehicles counting for 95.06% violate the speed limitation, a total of 15423 vehicles counting for 84.46% violate the following distance limitation, a total of 169 vehicles counting for 4.04% violate the drive on lane line limitation, and a total of 718 vehicles counting for 19.25% violate the lane-change limitation. Because of the slight congestion on the road, it is difficult for most vehicles to satisfy the minimum speed and distance requirements, resulting in a large proportion of violations. When making lane-change and overtaking, fewer vehicles violate the laws. Approximately 4.04% of vehicles drive on lane lines for over 6 s, which may be caused by a curved road, making it difficult to drive in the lane. Out of 718 instances of lane-change violations, there are 704 vehicles that keep an insufficient distance with RVTL because aggressive drivers cannot maintain a sufficient safe distance from the RVTL when making lane-change.

Statistical violation results for the 25th fragment in the AD4CHE dataset and typical illegal examples are shown in Fig.8b. This fragment lasted 290 s and contains 786 trajectories. The statistics for each type of violation were counted at intervals of 5 s. Among them, vehicle 9629 ran in the second inner lane at a speed of 55.5 km/h (far lower than the minimum speed limitation of 100 km/h)

with a following distance of 19.7 m (far shorter than the minimum compliance following distance of 50 m). Vehicle 9929 violated the lane-change article because of its short distance from the RVTL according to the compliance threshold in Equation 1. Vehicle 10053 drove on the lane line for approximately 7.4 s, which exceeded the compliance threshold of 6 s.

For the SIND dataset, in vehicles with the corresponding trigger condition activated, there are a total of 1288 vehicles counting for 9.91% violating the traffic light limitation, and a total of 101 vehicles counting for 1.85% violate the right of way limitation. More detailed statistics on different types of violations related to traffic lights in different regions can be found in Supplementary Result. The statistical results indicated that Chinese driver compliance with traffic laws was directly proportional to safety and punishment for violations. The statistical violation results for the 8_2_1 fragment in the SIND dataset are presented in Fig.8c. This fragment lasted 1,200 s and contained 611 trajectories. The statistics of the intersection violations for each type at intervals of 20 s are presented. Among them, vehicle 78 violated the right-of-way with a TTI_{diff} of 3.2 s, interfering with the straight movement of vehicle 73. Vehicle 365 violated the traffic light law, it ran the yellow light after the yellow light turned on 0.93 s ago. Vehicle 481 violated the traffic light laws. It crossed the stop line when the red light was on.”

We also found many interesting details when selecting thresholds and analyzing results. Due to the limitation of article length, we have placed these analyses in Supplementary Information-Some interesting findings. “Due to the lack of effective continuous monitoring, most rule violations get penalties only when incidents occur. Consequently, on highways, many drivers tend to prioritize efficiency over compliance. For instance, highway regulations stipulate a minimum 50 m following distance between vehicles, but the dataset reveals that most drivers fail to maintain this distance. Moreover, since rear-end collisions typically hold the rear vehicle responsible, a large majority of drivers pay less attention to maintaining distance from the RVTL when making lane-change. Among vehicles engaged in lane-change violations, most drivers are unable to maintain the safe distance from RVTL, even exhibiting highly aggressive cut-in behavior with a distance as little as 0.1 times the theoretical RSS distance.

However, as indicated in Fig.5c in the main paper, even exceptionally aggressive drivers will maintain a certain distance from RVTL during lane-changes, based on relative velocity. Below this threshold, all vehicles will be engaged in other driving behaviors. Furthermore, since most vehicles rarely perform significant deceleration during high-speed driving, drivers tend to reduce the distance to the leading vehicle. This might explain why some aggressive drivers only make rapid lane changes when their time gap to the leading vehicle falls below 2 seconds. This is also the reason why most

RVTLs do not exhibit noticeable avoidance behavior, even when confronted with dangerously close cut-in.

Fig. 8 Results of dataset validation. a, Statistical online monitoring results of dataset validation. b, Statistical online monitoring results and typical illegal examples for the 25th fragment in the AD4CHE dataset. c, Statistical online monitoring results and typical illegal examples for the 8_2_1 fragment in the SIND dataset.

Fig. 5c Different RVTL driving behaviors when the trigger is first activated

Fig. 6a Times to Intersection point (TTI) of left-turn and straight-moving vehicles

Table 3 Traffic light violations in different cities

City	Tianjin		Changchun		Chongqing		Xi'an	
Number of total monitoring instances	3433		4775		1379		3407	
Number and percentage of on stop line at the red light	202	5.88%	116	2.43%	62	4.5%	91	2.67%
Number and percentage of on stop line at the yellow light	13	0.38%	16	0.34%	7	0.51%	18	0.53%
Number and percentage of running the red light	42	1.22%	423	8.86%	10	0.73%	70	2.05%
Number and percentage of running the yellow light	111	3.23%	63	1.32%	30	2.18%	14	0.41%

At the intersections, as seen in Fig.6a in the main paper, it is evident that in left-turn vehicle passes first cases, left-turning vehicles arrive at the intersection mostly 2 seconds ahead of straight-moving vehicles. Instances of left-turning vehicles rush typically occur when left-turning vehicles are around 4-8 s away from the intersection point. Conversely, left-turning vehicles yielding the

straight-moving vehicles are commonly observed when they are 2-5 seconds away from the intersection points. Examining the points where the curves intersect the axes, vehicles generally pass the intersection points with a minimum time gap of around 2 s.

Supplementary Table 3 shows a more detailed result about the violations of traffic light. It can be observed that there are more instances of red-light violations than yellow-light violations. This might be due to the very short duration of yellow lights (approximately 3-5 s). If we consider violations on a time-based scale, there will be more vehicles running yellow lights per second. This is probably because the penalties for running red lights are more severe, and in some cities, running yellow lights goes unpunished. As a result, there is a noticeable regional disparity, as seen in Tianjin, where there are 111 instances of running yellow lights, roughly 2.6 times the number of red-light violations. This clearly indicates a lack of penalties for running yellow lights in Tianjin, whereas in Xi'an, there are only 14 instances of running yellow lights, approximately 0.2 times the number of red-light violations, indicating stricter enforcement in Xi'an. In the case of the intersection in Changchun, with only 7 data segments, a total of 423 vehicles were found to run red lights, comprising 77.6% of all red-light violations among the cities analyzed. This might be attributed to more aggressive driving culture in Changchun. Therefore, even though national regulations have been issued, there may be slight variations in the specific enforcement of these regulations in different regions. This leads to variations in driving behaviors among drivers in different areas, and it is a challenge that might need to be addressed in future compliance monitoring.”

Other significant comments 1:

The authors should significantly expand their literature review and better situate their contribution.

Response:

We have expanded our literature review and updated the contribution according to the reviewer's suggestions as explained in response to comment 1.

Other significant comments 2:

The authors should better explain why those chose the 5 specific traffic laws they did.

Response:

We agree with reviewer that why these specific traffic laws should be explained in detail. In accordance with the constraints imposed by traffic laws on driving behavior, we categorized these constraints into four types: vehicle speed, distance, driving action, and right-of-way. The English version of *Regulation on the Implementation of the Law of the People's Republic of China on Road*

Traffic Safety can be found at our figshare\Code\Supplementary Data\Original traffic law for further detail. Simultaneously, based on whether the constraints are continuous state constraints and if there is sufficient data in the dataset to support threshold analysis, we have selected six regulations to ensure they can encompass all types of constraints and are adequately supported by datasets. The relevant content is presented as following: “Traffic laws in most countries and regions limit driving behavior primarily through the following four aspects: vehicle speed, distance, driving action, and right-of-way as shown in Fig.2. Consider the *Regulation on the Implementation of the Law of the People’s Republic of China on Road Traffic Safety* [36] as an example. In Chapter 4 (“Road Traffic Regulations”), there are 49 traffic articles, but only 25 are related to motor vehicle driving behaviors. These 25 articles are subdivided into 93 articles, including 23 articles related to speed limits, 7 articles related to distance restrictions, 33 articles related to maneuvering restrictions, and 18 articles related to right-of-way. The detailed classification is shown in Fig.2.

Fig. 2 Classification of traffic laws. Each circle represents a different class of traffic-law constraint. The size of the circle represents the proportion of the class, and overlap indicates that both constraint classes appear in the same article. The solid edge line type of the circle indicates that there is no ambiguous expression involved, whereas the dash-dot line indicates that an ambiguous parameter is involved for certain articles and the dotted line indicates that the corresponding expressions in each article need to be clarified.

Six representative articles were selected that satisfied the requirements. i.e., they 1) included all constraint types, 2) involved both precise and ambiguous thresholds, 3) involved real-time and continuous state constraints, and 4) could be analyzed and validated using the available datasets. Four articles on highway regulations were selected according to these requirements.

Article 44: “The vehicle that intends to switch to another vehicle lane may do so on condition that it does not impede the normal running of other vehicles in the relevant lanes.”

Article 78: “When running on highway ..., where there are two vehicle lanes in the same direction, the minimum speed for the left lane is 100 kilometers per hour..., where there is any discrepancy between the speed indicated by a speed limit sign put up on a road and the driving speeds mentioned

above, a motor vehicle shall be driven at the speed indicated by the speed limit sign on the road.”

Article 80: “... when the speed is lower than 100 kilometers per hour, the distance from the vehicle in front may be narrowed appropriately, but the minimum distance may not be less than 50 meters.”

Article 82.6: “When driving a motor vehicle on the highway, the driver shall not drive over or on the dividing line of vehicle lanes or on the shoulder.”

The thresholds for the speed limit and following distance are specified precisely. However, the “*not impede*” and “*drive over or on the dividing line*” behaviors are ambiguous and need to be defined.

Two key articles on intersection regulations were selected.

Article 38.1: “When the green light is on, the vehicles are allowed to pass; when the yellow light is on, the vehicles that have gone beyond the stop line may continue to pass; when the red light is on, the vehicles are prohibited to pass.”

Article 38.2: “The vehicles making a turn may not interfere the vehicles that are let go straight forward.”

Article 38.1 involves both real-time and continuous state behavior constraints, which are precisely defined.

However, the “*not interfere*” behavior regarding right-of-way is ambiguous.”

Other significant comments 3:

The authors should better introduce the data sets and their limitations.

Response:

In the original submission, we only used partial AD4CHE data fragments. In this revision, we updated the monitor method and used all the data fragments to ensure sufficient data volume and coverage. The available data has been enlarged from 15 data fragments to 68 data fragments. In detail, the amount of instances including lane-changing behavior increased from 829 to 1753, the amount of instances used for analyzing TTC_{cl_min} increased from 328 to 1015, and the amount of instances used for analyzing d_{cl_min} increased from 150 to 1109.

In the original version of the SIND dataset, it only included one intersection data at Tianjin city. In the revised revision, we collected and calibrated more intersection datasets from three other cities (Changchun, Xi'an, Chongqing), added 30 new data fragments, approximately 537mins. Now, the SIND dataset covers cities spanning a range of 2300km, with high regional coverage. We also added the introduction of dataset in Datasets section: “We utilized two datasets for verification: The

AD4CHE (Aerial Dataset for China Congested Highway and Expressway) dataset [37] and the SIND (Signalized Intersection Dataset) dataset [38]. AD4CHE lasted approximately 307 mins, covering a trajectory length of 6,540.7 km and encompassing 53,761 trajectories, including 68 records captured on four Chinese highways. Compared with the HighD dataset [39], AD4CHE covers intricate road structures, including curved roads, on/off ramps, multiple lanes, and various traffic flow states, as shown in Figs.3(a–d), with abundant vehicle coordinate system parameters. In addition, we collected vehicle trajectories and traffic signal status information from four urban intersections to create the SIND dataset. This dataset encompassed four two-phase signalized intersections situated in different Chinese cities spanning 2,300 km. The SIND dataset lasted for approximately 957 mins, encompassing 30,953 trajectories, including 53 records and involving 7 types of traffic participants (cars, buses, trucks, motorcycles, bicycles, tricycles, and pedestrians), as shown in Figs.3(e–h). All available data were utilized to validate the online monitor. For further details, please refer to the source data.”

Fig. 3 Illustration figures of the Datasets. **a–d**, Road structure in AD4CHE [37]. **a**, Curved road. **b**, Curved road with export and import. **c**, Straight road with export and import. **d**, Straight road. **e–h**, Intersections in SIND [38]. **e**, Intersection in Changchun (43.88°N, terrain of low hills with a population of 9.05 million). **f**, Intersection in Tianjin (39.08°N, low-lying terrain of coastal plains with a population of 13.63 million). **g**, Intersection in Xi’an (34.15°N, terrain of plains and hills with a population of 12.99 million). **h**, Intersection in Chongqing (29.35°N, mountainous and hilly terrain with a population of 32.13 million).

[37] Zhang, Y. et al. The AD4CHE dataset and its application in typical congestion scenarios of traffic jam pilot systems. *IEEE Transactions on Intelligent Vehicles* (2023).

[38] Xu, Y. et al. SIND: A drone dataset at signalized intersection in China. *2022 IEEE 25th International Conference on Intelligent Transportation Systems (ITSC)* (2022).

[39] Krajewski, R., Bock, J., Kloeker, L. & Eckstein, L. The HighD dataset: A drone dataset of naturalistic vehicle trajectories on german highways for validation of highly automated driving systems. *21st international conference on intelligent transportation systems (ITSC)* (2018).

Other significant comments 4:

Some of the traffic laws require significant research and analysis to set the compliance threshold and the authors make interesting and valiant attempts to do so. This is novel work, as there is little existing precedent. I think the authors made many interesting observations in this section, but they

could set some more general principles and look for existing precedents (for example, the international regulations UNECE R157 and R79 set out parameters for following to closely, when lane changes interfere with cars in target lane, and how long lane changes should last, all of which this authors touch on without reference.

Response:

We agree with the reviewer that the compliance threshold was emphasized in the revised version, including the selection principles, the involved dataset, the trajectory filtering methods, and the calculations are all stated in detail in the Thresholds Analysis section. According to the suggestion of the reviewer, two general principles to determine the ambiguous compliance thresholds was proposed in the revised version. 1) *No Crashes*: For safety, there should be no crashes with other vehicles. This is ensured through safety-related indices such as the Time to Collision, Risk Sensitive Safety and other kinematics models. 2) *No Changes*: The ego vehicle's behavior should not be the reason to cause the change of other vehicles' behavior, which can be established by assessing nearby vehicles whose trajectories intersect with the ego vehicle. If no significant braking or steering responses are observed, we assume that the surrounding vehicle is not affected by the ego vehicle.

Specifically, according to the selected articles, there are four thresholds in ambiguous expressions that need to be determined: 1) the maximum allowable time to drive on the lane line (t_{cl_max}) was determined to specify the expression “*drive over or on the dividing line*” in Article 82.6; 2) when making lane-change, the minimum allowable TTC with the preceding vehicle (TTC_{cl_min}) and 3) the minimum allowable distance from the rear vehicle in the target lane (d_{cl_min}) were determined to specify the expression “*not impede*” in Article 44; 4) the minimum allowable time difference between a left-turn vehicle and a straight-moving vehicle to the intersection point (TTI_{diff_min}) was determined to specify the expression “*not interfere*” in Article 38.2.

When determine the threshold t_{cl_max} , TTC_{cl_min} , and d_{cl_min} , we did look for existing precedents such as regulations UNECE R157 and R79 as the references.

1) The maximum allowable time to drive on the lane line t_{cl_max} was determined utilizing the AD4CHE dataset with the *No Crashes* principle. Similar to the lane change maneuver (LCM) defined in the regulation UN ECE R157 [40], as shown in Fig.4a, the trigger condition for monitoring is that the ego vehicle's bounding box overlaps with the lane lines. In the dataset, 3,510 instances were recorded in which a vehicle implemented a lane-change. Among them, 1,753 instances involved complete lane-change trajectories. According to the collected data, the duration of all types of vehicles crossing lane lines during a lane-change was statistically analyzed, and the statistical results are shown in Fig.4c. The statistical results followed an inverse Gaussian

distribution with fitted parameters of $\mu=2.791$ and $\lambda=20.689$. Vehicles crossed lane lines for durations up to 6 s in 99.04% of the cases. Therefore, t_{cl_max} was determined to be 6 s, ensuring that standard lane-change maneuvers occurred within this specified time. This result differs from that in regulation UN ECE R79 [41], as t_{cl_max} is defined as the time when a vehicle overlaps the lane lines, rather than the entire lane-changing duration.

More specifically, as we employ fact-based monitoring, compliance is determined solely through vehicle behavioral data. Therefore, we commence the timing only when a vehicle overlaps the lane lines (as shown in Fig.4a). This differs from the definition of lane-changing time in UN ECE R79 section 5.6.4.6.5: “The lane change manoeuvre shall be completed in less than: (a) 5 seconds for M1, N1 vehicle categories; (b) 10 seconds for M2, M3, N2, N3 vehicle categories.” Therefore, the t_{cl_max} is slightly differed from the concept of time of lane change maneuver in UN ECE R79.

Fig. 4 Threshold analysis of the maximum allowable time to drive on the lane line t_{cl_max} and the minimum allowable TTC with the preceding vehicle TTC_{cl_min} . **a**, Trigger condition activation period. The trigger is active when the ego vehicle’s bounding box overlaps with the lane lines. **b**, TTC_{ini} with the preceding vehicle. Once the trigger is active, the TTC with the preceding vehicle is defined as TTC_{ini} . **c**, Distribution of the time to drive on the lane line. **d**, The ratio of t_{cl} to TTC_{ini} in different TTC_{ini} data points and fitting curve.

2) The minimum allowable TTC with the preceding vehicle TTC_{cl_min} was determined utilizing the AD4CHE dataset to resolve the “not impede” issue with preceding vehicles using the *No Crashes* principle. A total of 1,015 instances out of 1,753 complete trajectory data were utilized that the preceding vehicle was slower than the ego vehicle when the ego vehicle made a lane-change. The trigger condition for initiating monitoring is that the bounding box of the ego vehicle overlaps the lane lines and there is a preceding vehicle, the TTC between the ego vehicle and the preceding vehicle is defined as the initial TTC (TTC_{ini}). Considering that during the LCM period, TTC_{ini} should not be less than its time to drive on the lane line (t_{cl}), there will be a high risk of collisions that “impede” the preceding vehicle without any additional significant actions (Fig.4b). TTC_{ini} and t_{cl} were calculated, and their ratio was plotted on the coordinate axis, as shown in Fig.4d, and fitted. The fitting curve represents the relationship between TTC_{ini} and t_{cl} that most drivers follow

during the lane-change period. The TTC_{ini} at the intersection point of the fitted curve and the limiting value is 1.8 s, indicating that most drivers maintained a TTC_{cl_min} of 1.8 s with the preceding vehicle during the LCM. This is more conservative than the value of 2 s in ECE R157 results, given that ECE R157 considers a deviation of 0.375 m from the lane center line as the starting point, whereas we utilize the moment of overlapping with the lane line as our starting point, and only a few vehicles can cross that gap by 0.2 s.

- 3) The minimum allowable distance from the Rear Vehicle in the Target Lane (RVTL) d_{cl_min} is determined utilizing the AD4CHE dataset to solve the “not impede” issue with the RVTL using the *No Crashes* and *No Changes* principles. The d_{cl_min} value was determined as shown in Fig.5h in the revised manuscript,

$$d_{cl_min} = \begin{cases} 50, & \Delta v < -10.7 \\ -3.4\Delta v + 13.6, & -10.7 \leq \Delta v \leq 4 \\ 0, & \Delta v > 4 \end{cases}$$

where Δv represents the speed difference in m/s between the ego vehicle and RVTL at each sample time when the trigger activated.

Fig. 5h Optimal threshold line and RVTL' $d_{cl}-\Delta\bar{v}$ data points marked with different compliance states when the trigger is first activated in selected instances.

To verify the rational of selected compliance threshold, one extra experiment was proceeded refer to UN ECE R79. The critical situation was defined clearly in UN ECE R79: “A situation is deemed to be critical when, at the time a lane change manoeuvre starts, an approaching vehicle in the target lane would have to decelerate at a higher level than $3m/s^2$, 0.4s seconds after the lane change manoeuvre has started, to ensure the distance between the two vehicles is never less than that which the lane change vehicle travels in 1 second.”. After calculation, most of the critical situations in AD4CHE dataset defined in UN ECE R79 were judged as violations (below the threshold line) by the determined optimal compliance threshold, as shown in the figure below. This aligns with our general principle for selecting compliance thresholds: “*No Crashes* and *No Changes*.” The “*No Crashes*” criterion is inherently tied to safety considerations. When a scenario is deemed critical, it signifies that there may be an underlying potential for a collision risk.

Fig.1 Critical situation in R79 and the compliance threshold line. $\Delta v = v_{\text{ego}} - v_{\text{rear}}$

Furthermore, the minimum distance from the rear vehicle in an adjacent lane in UN ECE R79 is 55 m, however, it is only a few vehicles can keep that distance in the dataset. Because most vehicles rarely apply significant braking at high speeds, drivers tend to reduce the distance between vehicles.

[40] United Nations Economic Commission for Europe (UNECE). Regulation no 157 of the economic commission for Europe of the United Nations (un/ece) — uniform provisions concerning the approval of vehicles with regards to automated lane keeping systems (2021).

[41] United Nations Economic Commission for Europe (UNECE). Regulation no 79 of the economic commission for Europe of the United Nations (un/ece) — uniform provisions concerning the approval of vehicles with regard to steering equipment (2018).

Overall comments:

I think the application of traffic laws to AVs is an important area and this manuscript makes contributions, particularly in the exercise of trying to practically implement a traffic law violation detector and running it on an actual dataset. The results, obtained by running the detectors on real-world traffic data, have the potential to be quite interesting.

Right now, the manuscript is much more focused on the underlying MTL methodology, which is less novel. To the extent this remains the focus of the paper, it may represent an incremental advance which belongs in a more specialized journal.

To publish in this journal, I would urge the authors to expand, center, and put more focus on the results obtained from real world data and their broader implications. This will also necessitate grappling more with the formalization choices and the sensitivity of results to those choices. Such an expansion, particularly if it is brought into contact with the broader literature on driving behavior and safety, has potential to be a significant contribution.

Response:

Thanks for the reviewer's meticulous comments and encouragement. In this round revision, we have reorganized our main contributions to place a greater emphasis on an expanded and centralized compliance threshold analysis. This analysis is based on extensive data validation and has been integrated into our monitoring system using real driving behavior datasets and vehicle field tests.

Specifically, we expanded our literature review and reframed our contributions and provided enhanced details for Threshold Analysis and Results sections. We have expanded the coverage of our SIND dataset to support threshold analysis, sensitivity analysis, and monitoring result validation by reacquiring intersection data from three additional cities in different regions. Furthermore, we have obtained interesting findings, which are included in the Supplementary Information. Lastly, we conducted a preliminary real-vehicle test to verify our online monitoring system's feasibility and real-time capability.

REVIEWERS' COMMENTS

Reviewer #1 (Remarks to the Author):

Thank you for comprehensively addressing my comments on the paper.

Reviewer #3 (Remarks to the Author):

The authors have done a commendable job in expanding and reworking their articles to focus more closely on the problems of developing online traffic law behavior monitoring, implemented it robustly for on-road data, showed results, and discussed a real-world implementation. I really appreciate their responsiveness and engagement with the comments from all the reviewers.

To me, this paper significantly advances the state of the art in this field, has significant implications for the design and regulation of vehicle automation, and will be an interesting and useful read for multiple audiences. I have a small number overarching comments and specific comments, with the aim of improving the paper for publication.

I thank the authors for this very interesting and, in my view, valuable contribution.

Overarching

I think the manuscript could benefit from editing for clarity, particularly in the Results section. The figures are very useful. Perhaps some summary tables in the section can help organize and present results that are not contained well in graphical form.

If I am understanding correctly, the results in Figure 8 seem to show that it vehicles frequently violate traffic laws. Is this true? If so, it seems worthy of discussion. For example, table 3 in the supplementary information seems to contain significant interesting information.

Minor comments

The manuscript uses both the terms SV and AV - is there a difference between the two?

I believe RSS stands for "Responsibility Sensitive Safety" (page 3)

The concept of a hierarchical model for traffic law violations is introduced in Censi 2019; an online real-time planner based on traffic laws is described in Xiao, Mehdipour, et al 2021 ("Rule-based optimal control for autonomous driving") - these may be worth reviewing.

For t_{cl} , how did the reviewers know the distribution was inverse Gaussian and not, say, lognormal?

Why is t_{cl} aligned with the 99% observation in the dataset and not 90% or 99.9%?

RESPONSES TO REVIEWERS' COMMENTS

Manuscript ID: NCOMMS-23-20905A

Manuscript Title: Online Legal Driving Behavior Monitoring for Self-driving Vehicles

Authors: Wenhao Yu[†], Chengxiang Zhao[†], Hong Wang[✉], Jiaxin Liu, Xiaohan Ma, Yingkai Yang, Jun Li, Weida Wang, Xiaosong Hu[✉], Ding Zhao

REVIEWER COMMENTS

Reviewer #1 (Remarks to the Author):

Thank you for comprehensively addressing my comments on the paper.

Reviewer #3 (Remarks to the Author):

The authors have done a commendable job in expanding and reworking their articles to focus more closely on the problems of developing online traffic law behavior monitoring, implemented it robustly for on-road data, showed results, and discussed a real-world implementation. I really appreciate their responsiveness and engagement with the comments from all the reviewers.

To me, this paper significantly advances the state of the art in this field, has significant implications for the design and regulation of vehicle automation, and will be an interesting and useful read for multiple audiences. I have a small number overarching comments and specific comments, with the aim of improving the paper for publication.

I thank the authors for this very interesting and, in my view, valuable contribution.

Overarching

I think the manuscript could benefit from editing for clarity, particularly in the Results section. The figures are very useful. Perhaps some summary tables in the section can help organize and present results that are not contained well in graphical form.

If I am understanding correctly, the results in Figure 8 seem to show that it vehicles frequently violate traffic laws. Is this true? If so, it seems worthy of discussion. For example, table 3 in the supplementary information seems to contain significant interesting information.

Minor comments

The manuscript uses both the terms SV and AV - is there a difference between the two?
no difference

I believe RSS stands for “Responsibility Sensitive Safety” (page 3)

The concept of a hierarchical model for traffic law violations is introduced in Censi 2019; an online real-time planner based on traffic laws is described in Xiao, Mehdipour, et al 2021 (“Rule-based optimal control for autonomous driving”) - these may be worth reviewing.

For t_{cl} , how did the reviewers know the distribution was inverse Gaussian and not, say, lognormal?

Why is t_{cl} aligned with the 99% observation in the dataset and not 90% or 99.9%?

Response to the reviewers' comments

The authors extend our sincere appreciation for the additional comments and suggestions provided, which have proven invaluable in enhancing the manuscript. Below, we offer a detailed point-to-point response to each comment. Responses are typically presented in blue following the corresponding comments from the reviewers, and citations from the manuscript are likewise indicated in blue. We are grateful for the reviewers' insightful comments.

Reviewer 1

Thank you for comprehensively addressing my comments on the paper.

Response:

Thank you for the reviewer's confirmation. Your meticulous review has been instrumental in guiding us through significant improvements in our submission.

Reviewer 3

The authors have done a commendable job in expanding and reworking their articles to focus more closely on the problems of developing online traffic law behavior monitoring, implemented it robustly for on-road data, showed results, and discussed a real-world implementation. I really appreciate their responsiveness and engagement with the comments from all the reviewers.

To me, this paper significantly advances the state of the art in this field, has significant implications for the design and regulation of vehicle automation, and will be an interesting and useful read for multiple audiences. I have a small number overarching comments and specific comments, with the aim of improving the paper for publication.

I thank the authors for this very interesting and, in my view, valuable contribution.

Response:

Thanks for the reviewer's positive feedback. Your thorough examination and insightful feedback have greatly contributed to the refinement of our work. We truly appreciate your dedication to ensuring the quality of our submission.

Comment 1:

I think the manuscript could benefit from editing for clarity, particularly in the Results section. The figures are very useful. Perhaps some summary tables in the section can help organize and present results that are not contained well in graphical form.

If I am understanding correctly, the results in Figure 8 seem to show that it vehicles frequently violate traffic laws. Is this true? If so, it seems worthy of discussion. For example, table 3 in the supplementary information seems to contain significant interesting information.

Response:

We agree with the reviewer and add one table to show all the statistical results of our monitor outputs to make it easier to understand.

Table 1 Statistic of traffic rule violations

Traffic law violation behaviors		Number of traffic law violation instances	Percentage of traffic law violation instances	Number of total monitoring instances
Speed violation		18017	95.06%	18953
Following distance violation		15423	84.46%	18261
Driving on lane line		169	4.04%	4181
Lane-change violation	Insufficient TTC with front vehicle & insufficient distance with RVTL	7	0.19%	3729
	Insufficient TTC with front vehicle	7	0.19%	3729
	Insufficient distance with RVTL	704	18.88%	3729
Traffic light violation	Run the yellow light	218	1.68%	12994
	Run the red light	545	4.19%	12994
	On stop line at the yellow light	54	0.42%	12994
	On stop line at the red light	471	3.62%	12994
Right-of-way violation		101	1.85%	5467

Furthermore, we added more discussion in Supplementary information regarding this table:

Among all monitoring instances regarding speed violations, because of the congestion on the road, about 95.06% of the vehicles cannot satisfy the minimum speed requirement while no vehicle is over speed. The same reasons accounted for 84.46% of the monitored vehicles failing to meet the minimum following distance requirements. Such violations indicate that under certain specific working conditions, vehicles will find it challenging to adhere to traffic laws defined for typical situations. Therefore, leveraging the superior data processing capabilities of self-driving vehicles, the subsequent development of more refined traffic laws tailored for self-driving might be a solution. This approach aims to ensure that vehicles have adequate guidelines for driving without violations even in uncommon working conditions.

Based on the data fragment length and the number of monitoring instances, we also calculated the number of vehicles violating speed and following distance regulations under various traffic densities, as illustrated in the figures below. This aligns with the common understanding that an increase in traffic volume leads to a rise in the number of vehicles violating speed and following distance regulations.

Fig.1 The relationship between speed violations and traffic volume

Fig.2 The relationship between following distance violations and traffic volume

Comment 2:

The manuscript uses both the terms SV and AV - is there a difference between the two?

Response:

In our previous version, SV (Self-Driving Vehicles) and AV (Autonomous Vehicles) were nearly synonymous concepts. In this revised version, we have opted to use SV to prevent any potential confusion, given that both terms essentially refer to the same concept. This choice aims to enhance clarity and eliminate any ambiguity that may arise from interchangeable use.

Comment 3:

I believe RSS stands for "Responsibility Sensitive Safety" (page 3).

Response:

We appreciate your diligence in addressing the oversight. The correction of RSS to "Responsibility Sensitive Safety" has been duly noted. Additionally, we acknowledge your effort in thoroughly reviewing the entire document to rectify typos and grammar errors. These corrections contribute to the overall clarity and professionalism of the manuscript.

Comment 4:

The concept of a hierarchical model for traffic law violations is introduced in Censi 2019; an online real-time planner based on traffic laws is described in Xiao, Mehdipour, et al 2021 (“Rule-based optimal control for autonomous driving”) - these may be worth reviewing.

Response:

We agree with the reviewer and have reviewed these worthy references in our manuscript.

“Censi et.al proposed the concept of a hierarchical model for traffic law violation by implementing liability, ethics and culture-aware behavior specification as Rulebooks [17]. The primary applications of formalized traffic laws include monitoring, control synthesis, and formal verification [18]. Monitoring refers to checking the current or recorded driving behaviors of SVs violate traffic laws [11, 19]. The control synthesis aims to solve a vehicle controller to plan an optimal trajectory within traffic laws restrictions [20–23]. Specifically, Xiao et. al implemented the traffic laws into an online real-time planner by specifying their priorities by constructing a priority structure called Total Order over eQuivalence classes (TORQ) [21].”

[17] Censi, A., et al. Liability, ethics, and culture-aware behavior specification using rulebooks. International Conference on Robotics and Automation (ICRA), 8536-8542. (2019).

[21] Xiao, W. et al. Rule-based optimal control for autonomous driving. Proceedings of the ACM/IEEE 12th International Conference on Cyber-Physical Systems, 143–154 (2021).

Comment 5:

For t_{cl} , how did the reviewers know the distribution was inverse Gaussian and not, say, lognormal?

Why is t_{cl} aligned with the 99% observation in the dataset and not 90% or 99.9%?

Response:

We did try five functions to fit the distribution, including the inverse Gaussian and lognormal distributions. The plots and data can be found as below. It’s quite hard to find differences between inverse Gaussian and other distributions. It is essential to note that despite various fitting results, the percentage of compliance vehicle numbers almost consistent when $t_{cl}=6$ s as shown in Table 1. The inverse Gaussian describes the distribution of the time a Brownian motion with positive drift takes to reach a fixed positive level. Although the movement of vehicles is not a Brownian motion, the t_{cl} statistically measures the time vehicles take to cross the lane line, that is, the time it takes to travel a relatively fixed distance. This measurement aligns more closely with the content described by the inverse Gaussian distribution. Hence, we chose the inverse Gaussian distribution for fitting the distribution of t_{cl} .

Table 1 Indexes of different fitting results of t_{cl}

Distribution fitting function	Mean squared error	Center of distribution	[0, 6] s percentage
Inverse gaussian	7.5534	2.7908	99.039%
Log-normal	7.5473	2.7816	99.038%
Birnbaum-Saunders	7.5542	2.6198	99.08%
Generalized extreme value	7.5263	2.2761	98.212%
Log-logistic	7.5187	2.7514	98.556%

Regarding why we chose the t_{cl} aligned with the 99% observation in our study, we carefully considered this decision to ensure a higher confidence level in our analysis. Firstly, we opted for a 99% confidence level, where the corresponding t_{cl} value is very close to the integer 6s. Secondly, at a 90% confidence level, the t_{cl} value is 4.15s, and at a 95% confidence level, it is 4.72s. Through dataset replay, we observed that within the interval of 4.15-6s, there are still many instances of lane-changing behaviors, particularly slower, simple lane changes. Therefore, choosing 6s is more appropriate. The t_{cl} value for a 99.9% confidence level is 7.69s. However, most lane-changing

maneuvers typically occur within 6 seconds. Instances where a lane change extends beyond 6 seconds often result from the following vehicle not yielding promptly, thereby causing the changing vehicle to briefly ride along the lane line. Consequently, it is not advisable to further expand the confidence level. Additionally, considering that threshold values are best rounded to integers, we selected the t_{cl} value corresponding to the 99% observation, which is 6s.